# Elliptical Attention

**Stefan K. Nielsen**[*]
FPT Software AI Center
Ha Noi, Vietnam
stefannvkp@fpt.com

**Laziz U. Abdullaev**[*]
Department of Mathematics
National University of Singapore
Singapore 119077, Singapore
laziz.abdullaev@u.nus.edu

**Rachel S.Y. Teo**
Department of Mathematics
National University of Singapore
Singapore 119077, Singapore
rachel.teo@u.nus.edu

**Tan M. Nguyen**
Department of Mathematics
National University of Singapore
Singapore 119077, Singapore
tanmn@nus.edu.sg

## Abstract

Pairwise dot-product self-attention is key to the success of transformers that achieve state-of-the-art performance across a variety of applications in language and vision. This dot-product self-attention computes attention weights among the input tokens using Euclidean distance, which makes the model prone to representation collapse and vulnerable to contaminated samples. In this paper, we propose using a Mahalanobis distance metric for computing the attention weights to stretch the underlying feature space in directions of high contextual relevance. In particular, we define a hyper-ellipsoidal neighborhood around each query to increase the attention weights of the tokens lying in the contextually important directions. We term this novel class of attention Elliptical Attention. Our Elliptical Attention provides two benefits: 1) reducing representation collapse, and 2) enhancing the model's robustness as Elliptical Attention pays more attention to contextually relevant information, rather than focusing on some small subset of informative features. We empirically demonstrate the advantages of Elliptical Attention over the baseline dot-product attention and state-of-the-art attention methods on various practical tasks, including object classification, image segmentation, and language modeling across different data modalities. The code is publicly available at https://github.com/stefvk/Elliptical-Attention.

## 1 Introduction

Attention mechanisms and transformers [82] have achieved state of the art performance across a wide variety of tasks in machine learning [27, 35, 75] and, in particular, within natural language processing [1, 2, 13, 65, 12], computer vision [16, 39, 78, 66, 62], and reinforcement learning [25, 5]. They have also demonstrated strong performance in knowledge transfer from pretraining tasks to various downstream tasks with weak or no supervision [63, 64, 15]. At the core of these models is the dot-product self-attention mechanism, which learns self-alignment between tokens in an input sequence by estimating the relative importance of each token with respect to all others. The mechanism then transforms each token into a weighted average of the feature representations of the other tokens with weights proportional to the learned importance scores. The relative importance scores capture contextual information among tokens and are key to the success of the transformer architecture [83, 76, 8, 59, 36, 55].

---

[*]Equal contribution. Correspondence to stefannvkp@fpt.com

38th Conference on Neural Information Processing Systems (NeurIPS 2024).

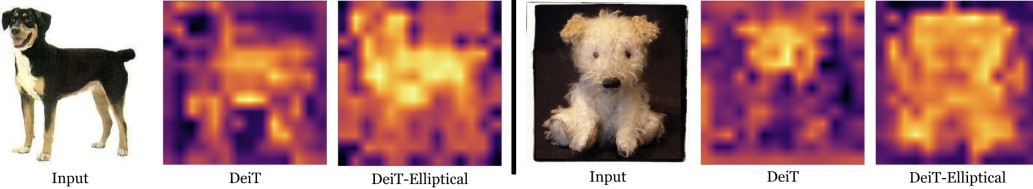

|  Input | DeiT | DeiT-Elliptical | Input | DeiT | DeiT-Elliptical |

Figure 1: Comparison of Attention Heatmaps. Elliptical pays attention to more relevant information. DeiT focuses on just a subset of informative features while Elliptical considers a wider set of contextually relevant information, helping to produce more accurate and robust predictions. Attention scores are min-max scaled for visualization purposes.

Recent work has begun exploring the connections between self-attention and non-parametric kernel regression [54, 23]. Under this interpretation, there is an unknown, underlying function $f$ mapping the tokens in the input sequence to the output sequence. The self-attention mechanism estimates $f$ by performing Nadaraya-Watson (NW) regression with isotropic Gaussian kernels. Our work leverages this perspective on self-attention, where we notice that Gaussian isotropic kernels are spherically invariant. This has the drawback of assuming all dimensions of the feature space are equal in terms of importance, meaning nearby tokens are assigned contextual relevance weights dependant only on their Euclidean distance from a query, regardless of direction. From the non-parametric regression perspective, we show that spherical invariance in the kernel causes the estimator to suffer provably higher variance. This causes two connected disadvantages in the self-attention setting. First, high variance in the estimator impairs robustness as small contaminations in the input cause large, erroneous changes in the self-attention output. Second, the high variance of the estimator reduces the capacity of the self-attention mechanism as hidden representations passing through the model are increasingly composed of uninformative noise.

**Contribution.** In this work, we propose Elliptical Attention, a new class of self-attention that constructs hyper-ellipsoidal, rather than hyper-spherical, neighborhoods around the attention queries. The key idea is to stretch the neighborhoods around the queries to upweight keys in directions of high importance. We achieve this by computing a Mahalanobis transformation that stretches the axes of the underlying feature space according to a learned measure of coordinate-wise relevance. Constructing hyper-ellipsoidal neighborhoods following this scheme allows the self-attention mechanism to learn higher-quality contextual representations that prevent representation collapse while simultaneously exhibiting stronger robustness. We additionally propose an estimator of coordinate-wise relevance in the self-attention mechanism that can be computed highly efficiently and with no learnable parameters. We theoretically prove that our estimator accurately estimates the relative coordinate-wise relevance in the feature space. Finally, our approach of constructing hyper-ellipsoidal neighborhoods is linked to theoretical improvements in the mean squared error (MSE) of non-parametric estimators by reducing variance without introducing bias. We demonstrate that this provable reduction in variance is related to both representation collapse and robustness, proposing a unifying framework for both phenomena. This framework is based on the geometry of the predictive neighborhood around queries in the attention mechanism. In summary, our contributions are three-fold:

1. We develop the novel Elliptical Attention, which learns better contextual representations by constructing hyper-ellipsoidal neighborhoods around queries.

2. We propose an efficient estimator of the coordinate-wise relevance in the self-attention mechanism, which requires no learnable parameters, and provide theoretical guarantees for this estimator.

3. We derive a theoretical framework unifying representation collapse and robustness in transformers based only on the implicit geometry of the attention mechanism.

We empirically demonstrate that 1) Elliptical Attention outperforms baseline self-attention models in terms of accuracy and robustness on a variety of practical benchmarks, including WikiText-103 language modelling, ImageNet-1K object classification, LRA long sequence modeling, and ADE20K image segmentation, 2) Elliptical Attention attains robust improvements with lower memory requirements and faster computational speed than baseline robust transformers, and 3) Elliptical Attention can be combined with state-of-the-art robust transformers to further boost robust performance in ImageNet-1K under adversarial attack.

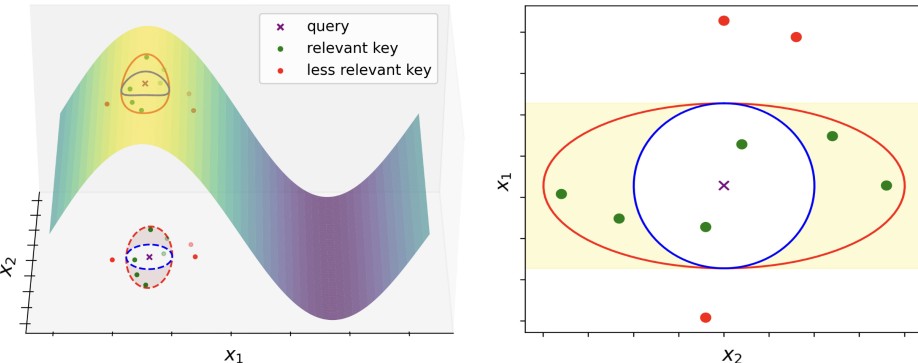

Figure 2: **Left:** The function does not vary in the $x_2$ axis so we stretch the neighborhood in that direction. **Right:** The stretched ellipsoidal neighborhood includes 4 more keys.

**Organization.** We structure this paper as follows: In Section 2, we present preliminaries on self-attention and non-parametric kernel regression. In Section 3, we illustrate the theoretical benefits of hyper-ellipsoidal neighborhoods, demonstrate how we build the required transformation, and provide the full technical formulation of Elliptical Attention. We empirically validate the advantages of the Elliptical Attention in Section 4. Related work is discussed in Section 5 before presenting concluding remarks in Section 6. Proofs, technical details, and further experiments are provided in the Appendix.

## 2 Background: Self-Attention and Non-Parametric Regression

We first provide preliminaries on the self-attention mechanism followed by background on its connection to the Nadaraya-Watson (NW) estimator in non-parametric regression [48].

### 2.1 Self-Attention Mechanism

Given an input sequence $\boldsymbol{X} = [\boldsymbol{x}_1, \ldots, \boldsymbol{x}_N]^\top \in \mathbb{R}^{N \times D_x}$ of $N$ feature vectors, the self-attention mechanism transforms the input to $\boldsymbol{H} := [\boldsymbol{h}_1, \ldots, \boldsymbol{h}_N]^\top \in \mathbb{R}^{N \times D_x}$ as follows:

$$\boldsymbol{h}_i = \sum_{j \in [N]} \text{softmax}\left(\frac{\boldsymbol{q}_i^\top \boldsymbol{k}_j}{\sqrt{D}}\right) \boldsymbol{v}_j, \text{ for } i = 1, \ldots, N. \tag{1}$$

The vectors $\boldsymbol{q}_i, \boldsymbol{k}_j$, and $\boldsymbol{v}_j$ are the query, key, and value vectors, respectively. They are computed as $[\boldsymbol{q}_1, \ldots, \boldsymbol{q}_N]^\top := \boldsymbol{Q} = \boldsymbol{X} \boldsymbol{W}_Q^\top \in \mathbb{R}^{N \times D}$, $[\boldsymbol{k}_1, \ldots, \boldsymbol{k}_N]^\top := \boldsymbol{K} = \boldsymbol{X} \boldsymbol{W}_K^\top \in \mathbb{R}^{N \times D}$, and $[\boldsymbol{v}_1, \ldots, \boldsymbol{v}_N]^\top := \boldsymbol{V} = \boldsymbol{X} \boldsymbol{W}_V^\top \in \mathbb{R}^{N \times D_v}$ where $\boldsymbol{W}_Q, \boldsymbol{W}_K \in \mathbb{R}^{D \times D_x}, \boldsymbol{W}_V \in \mathbb{R}^{D_v \times D_x}$ are the weight matrices. Eqn. 1 can be expressed in matrix form as:

$$\boldsymbol{H} = \text{softmax}\left(\frac{\boldsymbol{Q} \boldsymbol{K}^\top}{\sqrt{D}}\right) \boldsymbol{V}, \tag{2}$$

where the softmax function is applied row-wise to the matrix $\boldsymbol{Q} \boldsymbol{K}^\top / \sqrt{D}$. We refer to transformers built with Eqn. 2 as standard transformers or just transformers.

### 2.2 A Non-Parametric Regression Perspective of Self-Attention

We now present the connection between self-attention as described in Eqn. 1 and non-parametric regression. We first assume key and value vectors $\{\boldsymbol{k}_j, \boldsymbol{v}_j\}_{j \in [N]}$ are obtained from the following data generating process:

$$\boldsymbol{v} = f(\boldsymbol{k}) + \boldsymbol{\epsilon}, \tag{3}$$

where $\boldsymbol{\epsilon}$ is random zero-mean noise $\mathbb{E}[\epsilon] = 0$, and $f$ is the unknown function to be estimated. We consider the random design setting where the keys $\{\boldsymbol{k}_j\}_{j \in [N]}$ are i.i.d samples drawn from the marginal distribution $p(\boldsymbol{k})$. We use $p(\boldsymbol{v}, \boldsymbol{k})$ to denote the joint distribution of pairs $(\boldsymbol{v}, \boldsymbol{k})$ as obtained according to Eqn. 3. At any given new query $\boldsymbol{q}$, we aim to estimate the unknown function $f(\boldsymbol{q})$.

The NW estimator is a non-parametric estimator of the unknown $f$ described by

$$f(\boldsymbol{k}) = \mathbb{E}[\boldsymbol{v}|\boldsymbol{k}] = \int_{\mathbb{R}^D} \boldsymbol{v} \cdot p(\boldsymbol{v}|\boldsymbol{k}) d\boldsymbol{v} = \int_{\mathbb{R}^D} \frac{\boldsymbol{v} \cdot p(\boldsymbol{v}, \boldsymbol{k})}{p(\boldsymbol{k})} d\boldsymbol{v}, \tag{4}$$

where we apply zero-mean noise for the first equality and the definitions of conditional expectation and density for the second and final. Then, it can be shown that by estimating the joint density $p(\boldsymbol{v}, \boldsymbol{k})$ and marginal density $p(\boldsymbol{k})$ using isotropic Gaussian kernels with bandwidth $\sigma$ and evaluating the NW estimator at a new query $\boldsymbol{q}_i$, we obtain

$$\hat{f}_\sigma(\boldsymbol{q}_i) = \frac{\sum_{j\in[N]} \boldsymbol{v}_j \exp\left(-\|\boldsymbol{q}_i - \boldsymbol{k}_j\|^2/2\sigma^2\right)}{\sum_{j\in[N]} \exp\left(-\|\boldsymbol{q}_i - \boldsymbol{k}_j\|^2/2\sigma^2\right)} \tag{5}$$

$$= \frac{\sum_{j\in[N]} \boldsymbol{v}_j \exp(\boldsymbol{q}_i^\top \boldsymbol{k}_j/\sigma^2)}{\sum_{j\in[N]} \exp(\boldsymbol{q}_i^\top \boldsymbol{k}_j/\sigma^2)} = \sum_{j\in[N]} \mathrm{softmax}(\boldsymbol{q}_i^\top \boldsymbol{k}_j/\sigma^2)\boldsymbol{v}_j, \tag{6}$$

where choosing $\sigma^2 = \sqrt{D}$ as the isotropic variance recovers the full attention mechanism. We present the full derivation in Appendix A.

**Limitation of self-attention.** We see in Eqn. 5 that standard self-attention computes the relative importance scores between queries and keys via Euclidean distance. Euclidean distances are spherically invariant and therefore fail to consider coordinate-wise significance in the feature space, meaning the proximity of $\boldsymbol{k_j}$ from $\boldsymbol{q_i}$ influences its contextual relevance equally regardless of direction.

# 3 Elliptical Attention: Leveraging Hyper-Ellipsoids to Pay More Attention Without Losing Focus

In this section, we first present how NW regression obtains a lower MSE by taking hyper-ellipsoidal neighborhoods around queries. We then construct the required hyper-ellipsoidal transformation via a Mahalanobis metric. We present the framework relating robustness and representation collapse to the geometry of the query neighborhoods and show how our proposed scheme offers improvements in both areas. We then provide an efficient estimator of the coordinate-wise relevance before finally giving the full technical formulation of Elliptical Attention. Technical details on the implementation procedure are in Appendix E.

## 3.1 Improving NW Regression with Hyper-Ellipsoids

Distance-based estimators, such as the NW estimator, can obtain a lower MSE by taking hyper-ellipsoidal neighborhoods around queries [29, 30]. The key idea is that we wish to stretch the axes of the underlying space in directions for which the true $f$ in Eqn. 3 varies least.

Figure 2 shows a situation in which $f$ does not vary equally in all directions. This is actually a limiting case in which the function is sparse in the $x_2$ direction. In the left sub-figure, we show the result of stretching the Euclidean circular neighborhoods around each query in the $x_2$ direction for which the function does not vary. The right sub-figure then shows how the resulting ellipse in the $x_2$ direction can include additional data points without adding additional bias into the model. It is a well-established result that the variance of non-parametric estimates at a point is inversely proportional to the number of samples in that point's neighborhood, as the additional samples smooth out the effect of noise. As a result, stretching the neighborhood, as shown in the right sub-figure, decreases the variance. Crucially, including these additional samples does

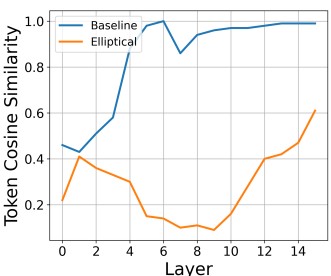

Figure 3: Representation Collapse on WikiText-103. Elliptical Attention learns more diverse representations.

not cause the estimate to miss the true variation in the function, as there is no variation in the $x_2$ direction. By including points in this direction, we do not introduce bias into the estimate. Hence, we lower variance without the introduction of bias, obtaining a lower MSE estimator. This intuition is formalized in Theorem 1 in Appendix C, which shows that the best achievable rate of convergence for estimators of non-sparse Lipschitz functions is of the order $\mathcal{O}(n^{-2/(2+d)})$ for a $d$ dimensional feature space. However, when the function only depends on $R \subseteq [d]$ coordinates, the rate improves to $\mathcal{O}(n^{-2/(2+|R|)})$. In the case of approximate sparsity, when coordinate directions exhibit differing variability, the same intuition carries over as shown by the improvement in convergence rates in Theorem 2 in Appendix C.

We leverage this analysis from non-parametric regression to motivate our Elliptical Attention. From the regression perspective, the self-attention mechanism, which performs NW regression, is able

to learn a lower MSE estimator of the true underlying $f$ by reducing the variance of the estimator without (or with minimal) introduction of bias. From the attention perspective, this means queries pay higher attention to more relevant keys, producing more contextually meaningful attention scores and better, more robust learned representations.

## 3.2 Capturing Coordinate-wise Variability and Building the Mahalanobis Transformation

We measure the variation in $f$ in the $i^{th}$ coordinate direction by the expectation of the $\mathcal{L}_1$ norm of the $i^{th}$ directional derivative taken over all $k \in \mathcal{X}_k$, where $\mathcal{X}_k \subseteq \mathbb{R}^D$ denotes the feature space. Roughly speaking, this quantity corresponds to the average absolute gradient of $f$ in the $i^{th}$ direction throughout the space. Formally, this quantity is defined as

**Definition 1 (Coordinate-wise Variability of $f : \mathbb{R}^D \to \mathbb{R}^{D_v}$)** *The coordinate-wise variability of $f : \mathbb{R}^D \to \mathbb{R}^{D_v}$ with Jacobian matrix $\boldsymbol{J}_f \in \mathbb{R}^{D_v \times D}$ in the $i^{th}$ direction is given by the quantity $\|f_i'\|_{1,\mu} := \mathbb{E}_{\boldsymbol{k} \sim \mu} \|\boldsymbol{J}_f(\boldsymbol{k})e_i\|_1, i \in [D]$, where $e_i$ is an all-zero vector with a single 1 in the $i^{th}$ coordinate and $\mu$ is the marginal distribution of $k$ over support $\mathcal{X}_k$.*

**Remark 1** *This definition is one of many possible. One could also take the supremum rather than the expectation or consider second derivatives. We select this definition as averages over first derivatives are more easily estimated and the definition still captures the intuitive properties of variability.*

Denoting estimates of the coordinate-wise variability $\|f_i'\|_{1,\mu}$ by $m_i$, we can then incorporate these quantities into a distance function of the form

$$d(\boldsymbol{q}, \boldsymbol{k}) := \sqrt{(\boldsymbol{q} - \boldsymbol{k})^\top \boldsymbol{M}(\boldsymbol{q} - \boldsymbol{k})}, \tag{7}$$

where $\boldsymbol{M} = \mathrm{diag}(m_1, m_2, \ldots, m_D)$ is a diagonal matrix whose diagonal elements are the estimates of $\|f_i'\|_{1,\mu}$ for $i \in [D]$.

**Remark 2** *The metric described in Eqn. 7 is a form of Mahalanobis distance metric, which can be interpreted as first applying a transformation to the underlying space in which we stretch the coordinate axes by the diagonal elements of $\boldsymbol{M}$. Therefore using this metric within the self-attention computation produces the desired hyper-ellipsoidal neighborhoods around queries.*

**Remark 3** *In practice, we maxscale the estimates to obtain $m_i \leftarrow m_i/m_{max}$ where $m_{max} \geq m_i$ for all $i \in [D]$. This is because we care about the relative magnitudes of the direction-wise variability as opposed to the absolute magnitudes. Under this interpretation, we identify the most variable dimension and stretch all others relative to this direction.*

## 3.3 Promoting Robustness and Avoiding Representation Collapse

Before providing the technical procedure for estimating $\boldsymbol{M}$ and the full technical formulation of Elliptical Attention in Section 3.5, we first theoretically analyze in Propositions 1 and 2 how the hyper-ellipsoidal transformation in Eqn.7 improves robustness and alleviates representation collapse.

**Dimension-wise input sensitivity of Elliptical Attention and robustness.** In Lemma 1, we show that when each input component is weighted according to the Mahalanobis transformation in Eqn. 7, the impact of perturbing the $i^{th}$ input coordinate on any coordinate of the output is proportional to the corresponding weighting parameter with proportionality coefficient depending on the indices $i$ and $j$.

**Lemma 1** *Let $\mathcal{M} : \mathbb{R}^D \to \mathbb{R}^N$ denote the transformed Elliptical* softmax *operator for a given set of keys as $\mathcal{M}(\boldsymbol{x}) := \frac{1}{\sum_{j \in [N]} \exp(\boldsymbol{x}^\top \boldsymbol{M} \boldsymbol{k}_j)} \left[ \exp(\boldsymbol{x}^\top \boldsymbol{M} \boldsymbol{k}_1), \exp(\boldsymbol{x}^\top \boldsymbol{M} \boldsymbol{k}_2), \ldots, \exp(\boldsymbol{x}^\top \boldsymbol{M} \boldsymbol{k}_N) \right]^\top$ for weight matrix $\boldsymbol{M}$ as in Eqn. 7. Then, the achievable rate of change of $\mathcal{M}(\boldsymbol{x})$ in $i^{th}$ input dimension is proportional to $m_i$, that is, $\sup_{\boldsymbol{x} \in \mathcal{X}} |\boldsymbol{J}_\mathcal{M}(\boldsymbol{x})_{ji}| \propto m_i$, for all $i \in [D]$ and $j \in [N]$ where $\boldsymbol{J}_\mathcal{M}$ is the Jacobian matrix of $\mathcal{M}$.*

By virtue of Lemma 1, which is proven in Appendix B.1, we show in Proposition 1 that choosing the weights as properly scaled estimates of the underlying function variability, as in Elliptical Attention, the output vectors become less prone to large errors caused by noisy input while simultaneously respecting the dimension-wise variability pattern of the true self-attention function.

**Proposition 1 (Robustness of Elliptical Attention)** *Let $f : \mathbb{R}^D \to \mathbb{R}^{D_v}$ be the true self-attention function, $\hat{f}_d$ be the Elliptical Attention estimator with metric $d$ as described in Eqn. 7. Then for any index $i \in [N]$ and noise $\epsilon \in \mathbb{R}^D$, the following error bound holds*

$$\|\hat{f}_d(\boldsymbol{q}_i) - \hat{f}_d(\boldsymbol{q}_i + \epsilon)\| \le \left( \sum_{j \in [N]} \sqrt{\text{tr}(\boldsymbol{K}_j^2 \boldsymbol{M}^2)} \|\boldsymbol{v}_j\| \right) \|\epsilon\|, \tag{8}$$

*where $\{\boldsymbol{K}_j\}_{j \in [N]}$ are constant diagonal matrices that depend only on the key vectors.*

Note that when the estimates are maxscaled so that $m_i \le 1$, the achievable output error of Elliptical Attention is lower than that of standard self-attention where $m_i = 1$ for all $i \in [D]$. Besides, when the true function exhibits approximate sparsity in some number of dimensions (i.e. $m_i \to 0^+$ for majority of indices), the error bound in Eqn. 8 becomes significantly tighter for Elliptical Attention. The proof of Proposition 1 is provided in Appendix B.2.

**Input smoothing and representation collapse.** In each layer, the standard self-attention mechanism fits a noisy estimate of the true function $f$, which is then fed into subsequent layers and iteratively refit. The input to each attention layer is then partially composed of noise, which is equivalently the common regularization method of random input smoothing. We show that by reducing the noise component in each layer, Elliptical Attention maintains expressive power and resists representation collapse. This is formalized in the following proposition:

**Proposition 2 (Elliptical Attention maintains expressive power by reducing noise)** *Let $\boldsymbol{h}_d^\ell$ denote the output of a transformer using Elliptical Attention with metric $d$ as described in Eqn. 7 and $\boldsymbol{h}^\ell$ denote the output of a transformer using standard self-attention at layer $\ell$. Let $\mathcal{D}$ be the sampling distribution of the data and let $\boldsymbol{c} \in \mathbb{R}^D$. Then, for any $\boldsymbol{h}, \boldsymbol{h}_d$ and layer $\ell$, in expectation a standard self-attention transformer attenuates towards $\boldsymbol{c}$ faster than Elliptical Attention. Formally, we have:*

$$\mathbb{E}_\mathcal{D} \|\boldsymbol{h}_d^\ell - \boldsymbol{c}\| \ge \mathbb{E}_\mathcal{D} \|\boldsymbol{h}^\ell - \boldsymbol{c}\|. \tag{9}$$

Proof is provided in Appendix B.3. Proposition 2 shows Elliptical Attention maintains better expressive power than standard self-attention. We find this empirically supported as shown in Fig 3.

### 3.4 An Efficient Estimator of the Coordinate-wise Variability

We propose a simple difference-based estimator that effectively captures the coordinate-wise variability of the underlying function. Our estimator is easily and efficiently computed. It requires no additional learnable parameters and demands negligible additional memory. Let $\mathbb{E}_n$ denote empirical mean over $n$ samples, $\boldsymbol{v}^\ell(i)$ denote the $i^{th}$ component of the vector $\boldsymbol{v}$ at the $\ell^{th}$ layer, and $\mathcal{X}_v^{\ell,\ell+1} = \{(\boldsymbol{v}^{\ell+1}, \boldsymbol{v}^\ell) : \boldsymbol{v}^\ell = f(\boldsymbol{k}^\ell) + \epsilon\}$ be the value feature space at neighboring layers $\ell$ and $\ell + 1$ where values are generated according to the process described in Eqn. 3. Then, our approach to estimating the $i^{th}$ coordinate-wise variability is described in the following proposition.

**Proposition 3 (Coordinate-wise Variability Estimator)** *Given a function $f : \mathbb{R}^D \to \mathbb{R}^{D_v}$ with $i^{th}$ directional variation $\|f_i'\|_{1,\mu}, i \in [D]$ and some $\delta > 0$, the directional variation can be estimated by the quantity*

$$m_i := \mathbb{E}_n \limits_{(\boldsymbol{v}^\ell, \boldsymbol{v}^{\ell+1}) \in \mathcal{X}_v^{\ell,\ell+1}} \frac{|\boldsymbol{v}^{\ell+1}(i) - \boldsymbol{v}^\ell(i)|}{\delta}. \tag{10}$$

**Remark 4** *For the purposes of improving the performance of transformers by stretching the feature space according to the direction-wise variability of $f$, we note that consistent estimators of $\|f_i'\|_{1,\mu}$ for all $i \in [D]$ are sufficient but not necessary. Instead, we require only the weaker objective of accurately estimating the relative magnitudes of the direction-wise variability. That is, if $\|f_i'\|_{1,\mu} \ge \|f_j'\|_{1,\mu}$, we need only that $m_i \ge m_j$. This is because the theory requires us only to identify coordinate directions of more or less variability and shrink or stretch the space accordingly.*

The intuition behind our estimator in Eqn. 10 lies in prior lines of research studying transformers as an Euler discretization of a continuous-time dynamic, usually as a system of first-order ordinary differential equations (ODEs) [40, 21, 53]. In fact, our estimator resembles the absolute value of a forward Euler discretization of the variability of the $i^{th}$ component of a value vector over time $\partial \boldsymbol{v}(i, t)/\partial t$, where the layers $\ell$ and $\ell + 1$ represent consecutive time points in an interval partition with the step size $\delta$. We prove that our estimator in Eqn. 10 effectively estimates the relative magnitudes of the coordinate-wise variability of $f$ in Appendix B.5.

## 3.5 Full Technical Formulation of Elliptical Attention

We now present the full formulation of Elliptical Attention. Given the distance function $d(\cdot, \cdot)$ as in Eqn. 7, where $\boldsymbol{M} = \mathrm{diag}(m_1, \ldots, m_D)$ is a diagonal matrix with elements $m_i$ as in Prop. 3, the $\boldsymbol{M}$-norm can be defined as $\|\boldsymbol{x}\|_{\boldsymbol{M}} := \sqrt{\boldsymbol{x}^T \boldsymbol{M} \boldsymbol{x}}$, which produces hyper-ellipsoidal stretching in the feature space. Then, Elliptical Attention is defined as follows.

**Definition 2 (Elliptical Attention Computation)** *Let* $\varphi_{d,\sigma} : \mathbb{R}^D \to \mathbb{R}$ *denote the Gaussian density kernel with variance* $\sigma^2 \boldsymbol{I}$ *equipped with the* $\boldsymbol{M}$-norm *as defined above. Then the corresponding NW estimator at* $\boldsymbol{q}_i$ *becomes*

$$\hat{f}_{d,D}(\boldsymbol{q}_i) := \frac{\sum_{j \in [N]} \boldsymbol{v}_j \exp\left(-\|\boldsymbol{q}_i - \boldsymbol{k}_j\|_{\boldsymbol{M}}^2 / 2\sigma^2\right)}{\sum_{j \in [N]} \exp\left(-\|\boldsymbol{q}_i - \boldsymbol{k}_j\|_{\boldsymbol{M}}^2 / 2\sigma^2\right)}. \tag{11}$$

*Then, the Elliptical Attention output for the* $i^{th}$ *query* $\boldsymbol{q}_i$ *given keys* $\{\boldsymbol{k}_i\}_{i=1}^N$ *and values* $\{\boldsymbol{v}_i\}_{i=1}^N$ *corresponding to the NW estimator (11) with* $\sigma^2 = \sqrt{D}$ *is given by*

$$\boldsymbol{h}_i = \sum_{j \in [N]} \frac{\exp\left(\boldsymbol{q}_i^\top \boldsymbol{M} \boldsymbol{k}_j / \sqrt{D}\right) \boldsymbol{v}_j}{\sum_{j \in [N]} \exp\left(\boldsymbol{q}_i^\top \boldsymbol{M} \boldsymbol{k}_j / \sqrt{D}\right)} = \sum_{j \in [N]} \mathrm{softmax}(\boldsymbol{q}_i^\top \boldsymbol{M} \boldsymbol{k}_j / \sqrt{D}) \boldsymbol{v}_j, \tag{12}$$

*where* $\boldsymbol{M} = \mathrm{diag}(m_1, \ldots, m_D)$ *with* $m_i$ *defined as Eqn. 10 for all* $i \in [D]$.

Eqn. 12 is equivalently expressed in matrix form as

$$\boldsymbol{H} = \mathrm{softmax}\left(\frac{\boldsymbol{Q}\boldsymbol{M}\boldsymbol{K}^\top}{\sqrt{D}}\right) \boldsymbol{V}. \tag{13}$$

**Remark 5** *We see from the form of Eqns. 12, 13 that standard self-attention is recoverable by setting* $\boldsymbol{M} = \boldsymbol{I}_D$. *Under our framework, this implies that standard self-attention assumes the underlying regression function to have exactly equal variability in all coordinate directions.*

Pseudocode for the Elliptical Attention computation is provided in Appendix F.12.

## 4 Experimental Results

In this section, we empirically justify the advantage of Elliptical Attention over baseline transformers that take hyper-spheres around queries. We evaluate our method on robust Wikitext-103 modeling under Word Swap contamination [45], ImageNet classification under a wide range of attacks [14, 67], the LRA benchmark [74], and ADE20K image segmentation [87]. We compare Elliptical Attention with state-of-the-art (SOTA) clean and robust models, including Performer [9], FourierFormer [54], Robust Vision Transformer [44], Fully Attentional Network (FAN) [89], Mixture of Gaussian Keys (MGK) [52], Mixture-of-Expert (MoE) based transformers, such as Switch transformer [18] and Generalist Langauge Model (GLaM) [17], and robust kernel density estimation (KDE) based transformers, such as Median of Means (MoM) and Scaled Projected KDE (SPKDE) [23]. We aim to show that i) Elliptical Attention offers substantive improvements over baseline models across tasks on both clean and contaminated data; ii) Elliptical Attention attains these improvements on contaminated data while reducing memory requirements and increasing computational speed compared to comparative robust models; iii) Elliptical Attention can be combined with SOTA robust transformers to further improve robustness with negligible increase in computational overhead. We compare Elliptical Attention with baselines of the same configuration. Results are averaged over 5 runs with different seeds. Additional results and full details on experimental setup are in Appendix F.

### 4.1 Robust Language Modelling

**Experimental setup.** We adopt the experimental setup in [54, 23]. We pretrain and evaluate our models on the WikiText-103 benchmark in comparison with the standard baseline Transformer [82], Performer [9], Transformer-MGK [52], FourierFormer [54], and the robust kernel density estimation-based Transformers including Transformer-SPKDE and Transformer-MoM [23]. All models use the 44M-parameter Transformer backbone. We pretrain all models on clean data for 125 epochs before attacking only the test set using a Word Swap Attack, which substitues random words with a generic 'AAA' token at a 2.5% swap rate. We report test perplexity (PPL) as the performance metric.

Table 1: Perplexity (PPL) on WikiText-103 under Word Swap contamination. Elliptical achieves top PPL in clean data and second best in contaminated. Best result in bold and second best underlined.

| Model | Clean Test PPL ($\downarrow$) | Contaminated Test PPL ($\downarrow$) |
|---|---|---|
| *Transformer* [82] | 34.29 | 74.56 |
| Performer [9] | 33.49 | 73.48 |
| Transformer-MGK [52] | 33.21 | 71.03 |
| FourierFormer [54] | 32.85 | 68.33 |
| Transformer-SPKDE [23] | 32.18 | 54.97 |
| Transformer-MoM [23] | 34.68 | **52.14** |
| Transformer-Elliptical | **32.00** | 52.59 |

Table 2: Top-1 and Top-5 Test accuracy on ImageNet under adversarial attacks PGD, FGSM, and SPSA with perturbation budget 1/255. Best result shown in bold and second best shown underlined.

| Method | Clean Data | | FGSM | | PGD | | SPSA | |
|---|---|---|---|---|---|---|---|---|
| | Top 1 | Top 5 | Top 1 | Top 5 | Top 1 | Top 5 | Top 1 | Top 5 |
| *DeiT* [78] | 72.23 | 91.13 | 52.61 | 82.26 | 41.84 | 76.49 | 48.34 | 79.36 |
| Distill [78] | 74.32 | 93.72 | 53.24 | 84.07 | 41.72 | 76.43 | 49.56 | 80.14 |
| FourierFormer [54] | 73.25 | 91.66 | 53.08 | 83.95 | 41.34 | 76.19 | 48.79 | 79.57 |
| RVT [44] | **74.37** | **93.89** | 53.67 | 84.11 | 43.39 | 77.26 | 51.43 | 80.98 |
| DeiT-KDE [23] | 72.58 | 91.34 | 52.25 | 81.52 | 41.38 | 76.41 | 48.61 | 79.68 |
| DeiT-MoM [23] | 71.94 | 91.08 | **55.76** | **85.23** | 43.78 | 78.85 | 49.38 | 80.02 |
| DeiT-Elliptical | 72.36 | 91.33 | 54.64 | 85.18 | **44.96** | **79.35** | **56.55** | **87.26** |

**Results.** Table 1 shows our Elliptical Transformer (*Elliptical*) achieves top test perplexity in clean data while also achieving second top test perplexity under data contamination by Word Swap [47], illustrating that the Elliptical Attention is highly robust and offers substantial advantages on clean data as well.

## 4.2 Image Classification under Adversarial Attack

**Experimental setup.** We adopt the experimental setup in [23]. We train and evaluate *Elliptical* on ImageNet-1K against standard vision transformers, including DeiT [78] and Distill [78], as well as the FourierFormer [54]. We also compare *Elliptical* with robust vision transformers, including DeiT-KDE [23], DeiT-MoM [23], RVT [44], and FAN [89]. The DeiT backbone is the tiny configuration of 5.7M parameters. We train all models on clean ImageNet-1K for 300 epochs before evaluating their top-1 and top-5 accuracy on the test dataset under fast gradient sign method (FGSM) [22], projected gradient descent (PGD) [42], and simultaneous perturbation stochastic approximation (SPSA) [81]. We also present results for performance against Auto Attack [11], which is an ensemble of auto PGD-Cross Entropy (APGD-CE), auto PGD-targeted (APGD-T), fast adaptive boundary-targeted (FAB-T), and Square. We display results for attacks individually and in default sequential mode.

**Results.** Table 2 shows *Elliptical* attains top robustness in PGD and SPSA and second top in FGSM while achieving highly competitive clean accuracy. *DeiT-Elliptical* is particularly impressive under black box attack SPSA, improving over the next best model, *RVT*, [44], by 10%. Table 4 shows results on Auto Attack [11], where we see *DeiT-Elliptical* substantially outperforms standard *DeiT* in each attack individually and sequentially. We again see strong performance against black box attack Square with an 8.5% improvement. When combining with SOTA robust transformer, *FAN* [89], Elliptical Attention improves robustness to sequential Auto Attack and all individual attacks except FAB-T, for which it still remains highly competitive. This shows Elliptical Attention can further boost robustness when combined with SOTA robust models.

## 4.3 Long Sequence Modelling on the LRA Benchmark

**Experimental setup.** We adopt the setup in [7]. For each of the 5 tasks, equation calculation (ListOps) [50], review classification (Text) [41], document retrieval (Retrieval) [61], image classification (Image) [32], and image spatial dependencies (Pathfinder) [37], we compare *Elliptical* with standard Transformer [82], Linformer [26], Reformer [28], Performer [9], and Longformer [3].

**Results.** Elliptical Attention achieves top or second top test accuracy in every task and top overall performance. This shows Elliptical Attention learns superior representations across a wide range of modalities in long-range contexts.

Table 3: Test accuracy on long range tasks: ListOps, Text, Retrieval, Image, and Pathfinder. Best result in bold and second best underlined.

| Dataset (seq. length) | *Trans.* [82] | *Lin.* [26] | *Re.* [28] | *Per.* [9] | *Long.* [3] | *Elliptical* |
|---|---|---|---|---|---|---|
| ListOps (2K) | 37.1 | 37.3 | 19.1 | 18.8 | 37.2 | **37.8** |
| Text (4K) | 65.0 | 55.9 | 64.9 | 63.8 | 64.6 | **65.6** |
| Retrieval (4K) | 79.4 | 79.4 | 78.6 | 78.6 | **81.0** | 80.3 |
| Image (1K) | 38.2 | 37.8 | **43.3** | 37.1 | 39.1 | 40.2 |
| Pathfinder (1K) | **74.2** | 67.6 | 69.4 | 69.9 | 73.0 | 73.2 |
| Average Accuracy | 58.5 | 55.6 | 55.1 | 53.6 | 59.0 | **59.4** |

Table 4: Top-1 and Top-5 Test accuracy on ImageNet under Auto Attack applied both individually and sequentially with perturbation budget 1/255. Best result is shown in bold.

| Method | *DeiT* [78] | | DeiT-Elliptical | | *FAN* [89] | | FAN-Elliptical | |
|---|---|---|---|---|---|---|---|---|
| | Top 1 | Top 5 | Top 1 | Top 5 | Top 1 | Top 5 | Top 1 | Top 5 |
| Clean Data | 72.23 | 91.13 | **72.36** | **91.33** | 76.31 | 93.42 | **76.38** | **93.53** |
| APGD-CE | 27.75 | 66.48 | **31.27** | **68.28** | 35.05 | 74.56 | **36.13** | **75.69** |
| APGD-T | 27.74 | 73.37 | **29.69** | **74.39** | 35.02 | 80.46 | **36.25** | **81.30** |
| FAB-T | 71.61 | 90.54 | **71.74** | **90.81** | 76.35 | 93.65 | 76.16 | 93.45 |
| Square | 43.55 | 80.96 | **47.25** | **81.65** | 56.75 | 88.05 | **58.38** | **88.20** |
| Average | 42.66 | 77.84 | **45.00** | **78.78** | 50.79 | 84.18 | **51.73** | **84.66** |
| Sequential Attack | 26.08 | 64.18 | **27.45** | **67.77** | 33.29 | 74.52 | **34.54** | **75.67** |

Table 5: Switch Transformer Language Modeling

| Model | Test PPL ($\downarrow$) |
|---|---|
| *Switch Transformer-medium* [18] | 35.33 |
| Switch Elliptical-medium | **34.67** |
| *Switch Transformer-large* [18] | 31.18 |
| Switch Elliptical-large | **30.56** |

Table 6: GLaM Language Modeling

| Model | Test PPL ($\downarrow$) |
|---|---|
| *GLAM-small* [17] | 58.27 |
| GLaM-Elliptical-small | **56.69** |
| *GLaM-medium* [17] | 38.27 |
| GLaM-Elliptical-medium | **36.34** |

## 4.4 Image Segmentation on ADE20K

**Experimental setup.** We adopt the setup in [71]. The encoder is pretrained on ImageNet-1K following the same specification described in 4.2. In particular, the encoder is a DeiT-tiny backbone of 5.7M parameters pretrained for 300 epochs. After pretraining, we then attach a decoder that contains 2-layer masked transformer and finetune the full encoder-decoder model for 64 epochs on the ADE20K [88] image segmentation dataset.

**Results.** Table 7 reports pixel accuracy, mean accuracy, and mean intersection over union (IOU). Elliptical Attention boosts performance across all metrics, with intersection over union, in particular, improving by a substantive 4.7%.

## 4.5 Further Clean Data Language Modelling

**Experimental setup.** For experiments using Switch Transformer [18] and GLaM [17] backbones, we adopt the setup in [60]. In particular, we integrate *Elliptical* into small (70M parameters) and medium (220M parameters) GlaM backbones and train the models on WikiText-103 for 80 and 120 epochs, respectively. We consider Switch backbones at medium (220M parameters) and large (388M parameters) configurations, both trained for 80 epochs. All models use top-2 expert routing. For the standard transformer experiments, we continue with the setup of [54] and additionally present results for *Elliptical* in a medium configuration with 90M parameters trained for 100 epochs.

**Results.** We present in Tables 5 and 6 the performance of *Elliptical* in MoE backbones. We see moving from smaller to larger configurations, *Elliptical* maintains strong, consistent improvements in test PPL. We note particularly substantive improvements with scale in the GLaM backbone, where at the small configuration *Elliptical* attains a 2.7% improvement, but at the medium configuration this performance improvement almost doubles to 5.0%. Table 8 further shows that in the standard transformer backbone, *Elliptical* maintains its substantive 6.8% improvement when scaling up to a larger configuration. These results show that Elliptical Attention scales well with model size.

| Table 7: Image Segmentation Results | | | |
|---|---|---|---|
| Model | Pixel Acc. | Avg Acc. | Avg IOU |
| *DeiT* [78] | 77.93 | 46.30 | 35.44 |
| Elliptical | **78.46** | **48.04** | **37.09** |

| Table 8: Wikitext-103 Results | |
|---|---|
| Model | Test PPL ($\downarrow$) |
| *Transformer-small* [82] | 34.29 |
| Elliptical-small | **32.00** |
| *Transformer-medium* [82] | 29.60 |
| Elliptical-medium | **27.60** |

## 5 Related Work

**Theoretical Frameworks for Attention.** Attention mechanisms have been studied from a range of perspectives. [80] shows that self-attention can be derived from kernel similarity functions, and [77] points out that self-attention projects its query vectors onto the principal component axes of its key matrix in a feature space. [56] formulates self-attention as the support vector expansion derived from a support vector regression problem, while [73] explains attention through nonlinear singular value decomposition of asymmetric kernels. Attention has also been explained through ordinary/partial differential equations, Gaussian mixture models, and graph-structured learning [40, 68, 51, 72, 20, 52, 31, 86]. [54, 23] show that self-attention performs Nadaraya-Watson regression with Gaussian isotropic kernels. This paper leverages this viewpoint and proposes modifying the Gaussian isotropic kernels to include a Mahalanobis metric which can be interpreted as stretching the hyper-spherical neighborhoods of the kernel to hyper-ellipsoids.

**Robust Transformers.** In vision, [43] proposes an ensemble defense strategy to white-box attacks while [44] proposes position-aware attention scaling and patch-wise augmentation. Recently, [89] proposes a fully-attentional network to attain state-of-the-art accuracy on corrupted image data. In language, [85] proposes structurally aware table-text encoding, [38] proposes a robust end-to-end transformer for crisis detection, and [33] proposes duration-based hard attention. [6, 4] integrate a Gaussian process into attention for out-of-distribution detection, and [79] develops equivariant neural functional networks for transformers. These methodologies are motivated by their respective domain and tend to have limited generalizability to differing domains. Our approach, by contrast, proposes a general framework that makes no assumption on the downstream task and requires no additional parameters and negligible computational overhead.

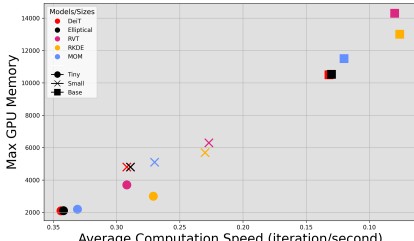

Figure 4: ImageNet Efficiency: Comparison of throughput and max memory allocated for DeiT, Elliptical, RVT, RKDE, MoM on Tiny, Small, and Base sizes. Elliptical is the most efficient robust model. Numerical analysis in Table 12 of Appendix F.

**Mahalanobis Metrics.** Mahalanobis metrics have been used predominantly in classical machine learning algorithms. In nearest-neighbor (NN) classification and regression, [84, 49] learn the metric through backpropagation. In NN KL divergence estimation, [58] learns a Mahalanobis metric from density approximation. In kernel regression, [57] takes eigenvalues of the estimated Jacobian while [29, 30] estimate coordinate-wise variability of the true function. Our model similarly uses coordinate-wise variability of the unknown function to form the Mahalanobis transformation but instead uses a more efficient estimator that does not require materializing the prediction function and accommodates the self-attention setting. In general, our method is among the early work in incorporating Mahalanobis metrics into the self-attention mechanism.

## 6 Conclusion and Future Work

In this paper, we present Elliptical Attention, a novel variant of attention that computes a Mahalanobis transformation to stretch the underlying feature space in directions of high contextual relevance. This transformation can be interpreted as modifying the hyper-spherical neighborhoods around queries to hyper-ellipsoids which upweight the attention paid to keys lying in important directions, enabling the transformer to learn better and more robust representations. This approach makes no assumptions on the downstream task, requires no learnable parameters, and can be applied to any transformer to boost clean and robust performance. A limitation of our work is that we use the values over layers to estimate the average direction-wise gradient of the true self-attention function, which makes the estimate prone to noise. For ongoing work, we are exploring more precise estimation methods with provable convergence guarantees that do not compromise efficiency.

## Acknowledgments and Disclosure of Funding

This research / project is supported by the National Research Foundation Singapore under the AI Singapore Programme (AISG Award No: AISG2-TC-2023-012-SGIL). This research / project is supported by the Ministry of Education, Singapore, under the Academic Research Fund Tier 1 (FY2023) (A-8002040-00-00, A-8002039-00-00). This research / project is also supported by the NUS Presidential Young Professorship Award (A-0009807-01-00).

Thanks to our anonymous reviewers, who provided valuable feedback which improved the paper substantially. Thanks also to Thai Ha for the many illuminating conversations.

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

# Supplement to "Elliptical Attention"

**Table of Contents**

## A  Full Derivation of Self-Attention as Non-Parametric Regression

Recall NW estimator is a non-parametric estimator of the unknown $f$ at any given query $\boldsymbol{q}$ described by

$$f(\boldsymbol{k}) = \mathbb{E}[\boldsymbol{v}|\boldsymbol{k}] = \int_{\mathbb{R}^D} \boldsymbol{v} \cdot p(\boldsymbol{v}|\boldsymbol{k}) d\boldsymbol{v} = \int_{\mathbb{R}^D} \frac{\boldsymbol{v} \cdot p(\boldsymbol{v}, \boldsymbol{k})}{p(\boldsymbol{k})} d\boldsymbol{v},$$

where the first equality comes from the noise being zero mean, the second equality comes from the definition of conditional expectation and the final equality comes from the definition of conditional density. Eqn. 3 implies that if we can just obtain good estimates of the joint density $p(\boldsymbol{v}, \boldsymbol{k})$ and marginal density $p(\boldsymbol{k})$ then we can estimate the required $f(\boldsymbol{q})$. The Gaussian isotropic kernels with

bandwidth $\sigma$ are given by

$$\hat{p}_\sigma(\boldsymbol{v}, \boldsymbol{k}) = \frac{1}{N} \sum_{j\in[N]} \varphi_\sigma(\boldsymbol{v} - \boldsymbol{v}_j)\varphi_\sigma(\boldsymbol{k} - \boldsymbol{k}_j), \;\; \hat{p}_\sigma(\boldsymbol{k}) = \frac{1}{N} \sum_{j\in[N]} \varphi_\sigma(\boldsymbol{k} - \boldsymbol{k}_j), \qquad (14)$$

where $\varphi_\sigma$ is the multivariate Gaussian density function with diagonal covariance matrix $\sigma^2 \boldsymbol{I}_D$. Given the kernel density estimators in Eqn. 14, the unknown function can be estimated as

$$\hat{f}_\sigma(\boldsymbol{k}) = \int_{\mathbb{R}^D} \frac{\boldsymbol{v} \cdot \hat{p}_\sigma(\boldsymbol{v}, \boldsymbol{k})}{\hat{p}_\sigma(\boldsymbol{k})} \, d\boldsymbol{v} = \int_{\mathbb{R}^D} \frac{\boldsymbol{v} \cdot \sum_{j\in[N]} \varphi_\sigma(\boldsymbol{v} - \boldsymbol{v}_j)\varphi_\sigma(\boldsymbol{k} - \boldsymbol{k}_j)}{\sum_{j\in[N]} \varphi_\sigma(\boldsymbol{k} - \boldsymbol{k}_j)} \, d\boldsymbol{v}$$

$$= \frac{\sum_{j\in[N]} \varphi_\sigma(\boldsymbol{k} - \boldsymbol{k}_j) \int \boldsymbol{v} \cdot \varphi_\sigma(\boldsymbol{v} - \boldsymbol{v}_j)d\boldsymbol{v}}{\sum_{j\in[N]} \varphi_\sigma(\boldsymbol{k} - \boldsymbol{k}_j)} = \frac{\sum_{j\in[N]} \boldsymbol{v}_j \varphi_\sigma(\boldsymbol{k} - \boldsymbol{k}_j)}{\sum_{j\in[N]} \varphi_\sigma(\boldsymbol{k} - \boldsymbol{k}_j)}.$$

Then, using the definition of the Gaussian isotropic kernel and evaluating the estimated function at $\boldsymbol{q}_i$ we have

$$\hat{f}(\boldsymbol{q}_i) = \frac{\sum_j^N \boldsymbol{v}_j \exp\left(-\|\boldsymbol{q}_i - \boldsymbol{k}_j\|^2/2\sigma^2\right)}{\sum_j^N \exp\left(-\|\boldsymbol{q}_i - \boldsymbol{k}_j\|^2/2\sigma^2\right)}$$

$$= \frac{\sum_j^N \boldsymbol{v}_j \exp\left[-(\|\boldsymbol{q}_i\|^2 + \|\boldsymbol{k}_j\|^2)/2\sigma^2\right] \exp(\boldsymbol{q}_i^\top \boldsymbol{k}_j/\sigma^2)}{\sum_j^N \exp\left[-(\|\boldsymbol{q}_i\|^2 + \|\boldsymbol{k}_j\|^2)/2\sigma^2\right] \exp(\boldsymbol{q}_i^\top \boldsymbol{k}_j/\sigma^2)}$$

$$= \frac{\sum_j^N \boldsymbol{v}_j \exp(\boldsymbol{q}_i^\top \boldsymbol{k}_j/\sigma^2)}{\sum_j^N \exp(\boldsymbol{q}_i^\top \boldsymbol{k}_j/\sigma^2)} = \sum_{j\in[N]} \mathrm{softmax}(\boldsymbol{q}_i^\top \boldsymbol{k}_j/\sigma^2)\boldsymbol{v}_j.$$

**Remark 6** *Note that relaxing the assumption of normalized keys, the standard unnormalized self-attention score can be written as*

$$\exp(\boldsymbol{q}_i^\top \boldsymbol{k}_j/\sigma^2) = \exp\left(-\|\boldsymbol{q}_i - \boldsymbol{k}_j\|^2/2\sigma^2\right) \exp\left((\|\boldsymbol{q}_i\|^2 + \|\boldsymbol{k}_j\|^2)/2\sigma^2\right)$$

$$\propto \exp\left(-\|\boldsymbol{q}_i - \boldsymbol{k}_j\|^2/2\sigma^2\right),$$

*which shows that the dot-product self-attention scores are proportional to the NW kernel value with Euclidean distance. Hence the assumption of key normalization is sufficient to recover exactly the correspondence between self-attention and NW kernel regression, but not necessary. Analogously, the unnormalized Elliptical Attention score takes the following form:*

$$\exp(\boldsymbol{q}_i^\top \boldsymbol{M}\boldsymbol{k}_j/\sigma^2) = \exp\left(-d(\boldsymbol{q}_i, \boldsymbol{k}_j)^2/2\sigma^2\right) \exp\left((\|\boldsymbol{q}_i\|_{\boldsymbol{M}}^2 + \|\boldsymbol{k}_j\|_{\boldsymbol{M}}^2)/2\sigma^2\right)$$

$$\propto \exp\left(-d(\boldsymbol{q}_i, \boldsymbol{k}_j)^2/2\sigma^2\right),$$

*where $d(\cdot, \cdot)$ is the Mahalanobis distance used in Eqn. 7 and $\|\cdot\|_M$ is the norm in the transformed space with metric $d$. This observation justifies the use of the transformed dot product instead of the full Mahalanobis distance metric in Eqn. 12 as it preserves the proportionality relationship between the attention computation and the corresponding nonparametric regression estimator with chosen distance metric.*

## B    Technical Proofs

In this section, we present the omitted theorem statements and technical proofs in the main body of the paper.

### B.1    Proof of Lemma 1

Let $\mathcal{M} : \mathbb{R}^D \to \mathbb{R}^N$ be the transformed $\mathrm{softmax}$ operator as defined in Lemma 1. We wish to find its Jacobian matrix given by

$$\boldsymbol{J}_\mathcal{M}(\boldsymbol{q}) = \begin{bmatrix} \frac{\partial \mathcal{M}_1(\boldsymbol{q})}{\partial q^1} & \frac{\partial \mathcal{M}_1(\boldsymbol{q})}{\partial q^2} & \cdots & \frac{\partial \mathcal{M}_1(\boldsymbol{q})}{\partial q^D} \\ \frac{\partial \mathcal{M}_2(\boldsymbol{q})}{\partial q^1} & \frac{\partial \mathcal{M}_2(\boldsymbol{q})}{\partial q^2} & \cdots & \frac{\partial \mathcal{M}_2(\boldsymbol{q})}{\partial q^D} \\ \vdots & \vdots & \ddots & \vdots \\ \frac{\partial \mathcal{M}_N(\boldsymbol{q})}{\partial q^1} & \frac{\partial \mathcal{M}_N(\boldsymbol{q})}{\partial q^2} & \cdots & \frac{\partial \mathcal{M}_N(\boldsymbol{q})}{\partial q^D} \end{bmatrix},$$

to measure the sensitivity of each output dimension to a change in each input dimension. Let $\mathcal{M}_j : \mathbb{R}^D \to \mathbb{R}$ denote the $j^{th}$ component of the output vector for $j \in [N]$, that is, for a vector $\boldsymbol{q} \in \mathbb{R}^D$,

$$\mathcal{M}_j(\boldsymbol{q}) = \frac{\exp(\boldsymbol{q}^\top \boldsymbol{M} \boldsymbol{k}_j)}{\sum_{s \in [N]} \exp(\boldsymbol{q}^\top \boldsymbol{M} \boldsymbol{k}_s)}. \tag{16}$$

Let $q^i$ and $k_j^i$ denote the $i^{th}$ coordinates of vectors $\boldsymbol{q}$ and $\boldsymbol{k}_j$, respectively. Then,

$$\frac{\partial}{\partial q^i} \ln(\mathcal{M}_j(\boldsymbol{q})) = \frac{\partial}{\partial q^i} \left( \boldsymbol{q}^\top \boldsymbol{M} \boldsymbol{k}_j - \ln \left( \sum_{s \in [N]} \exp(\boldsymbol{q}^\top \boldsymbol{M} \boldsymbol{k}_s) \right) \right)$$

$$= m_i k_j^i - \frac{\sum_{s \in [N]} \frac{\partial}{\partial q^i} \exp(\boldsymbol{q}^\top \boldsymbol{M} \boldsymbol{k}_s)}{\sum_{s \in [N]} \exp(\boldsymbol{q}^\top \boldsymbol{M} \boldsymbol{k}_s)}$$

$$= m_i k_j^i - m_i \sum_{s \in [N]} \frac{k_s^i \exp(\boldsymbol{q}^\top \boldsymbol{M} \boldsymbol{k}_s)}{\sum_{s' \in [N]} \exp(\boldsymbol{q}^\top \boldsymbol{M} \boldsymbol{k}_{s'})}$$

$$= m_i \left( k_j^i - \sum_{s \in [N]} k_s^i \mathcal{M}_s(\boldsymbol{q}) \right).$$

Since the output of Eqn. 16 consists of only positive components, we have

$$\frac{\partial}{\partial q^i} \mathcal{M}_j(\boldsymbol{q}) = \frac{\partial}{\partial q^i} \ln(\mathcal{M}_j(\boldsymbol{q})) \cdot \mathcal{M}_j(\boldsymbol{q})$$

$$= m_i \left( k_j^i - \sum_{s \in [N]} k_s^i \mathcal{M}_s(\boldsymbol{q}) \right) \mathcal{M}_j(\boldsymbol{q}).$$

Therefore, the triangle inequality gives

$$\left| \frac{\partial}{\partial q^i} \mathcal{M}_j(\boldsymbol{q}) \right| = \left| m_i \left( k_j^i - \sum_{s \in [N]} k_s^i \mathcal{M}_s(\boldsymbol{q}) \right) \mathcal{M}_j(\boldsymbol{q}) \right|$$

$$\leq m_i \left( |k_j^i(1 - \mathcal{M}_j(\boldsymbol{q})) \mathcal{M}_j(\boldsymbol{q})| + \sum_{s \in [N] \setminus \{j\}} |k_s^i \mathcal{M}_s(\boldsymbol{q}) \mathcal{M}_j(\boldsymbol{q})| \right). \tag{17}$$

We now bound each term individually. Consider the terms $j \neq s$ first. Since $0 \leq \mathcal{M}_s(\boldsymbol{q}) \leq 1$, we can bound them as

$$|k_s^i \mathcal{M}_s(\boldsymbol{q}) \mathcal{M}_j(\boldsymbol{q})| \leq |k_s^i|. \tag{18}$$

Now recall that the inequality $ab \leq (a+b)^2/4$ holds for any real numbers $a$ and $b$ with equality holding at $a = b$. Therefore, for the first term, we obtain

$$|k_j^i(1 - \mathcal{M}_j(\boldsymbol{q})) \mathcal{M}_j(\boldsymbol{q})| \leq |k_j^i| \frac{(1 - \mathcal{M}_j(\boldsymbol{q}) + \mathcal{M}_j(\boldsymbol{q}))^2}{4} = \frac{|k_j^i|}{4}. \tag{19}$$

Combining inequalities 17, 18 and 19, we finally arrive at

$$|\boldsymbol{J}_{\mathcal{M}}(\boldsymbol{q})_{ji}| = \left| \frac{\partial}{\partial q^i} \mathcal{M}_j(\boldsymbol{q}) \right| \leq m_i \left( \frac{|k_j^i|}{4} + \sum_{s \in [N] \setminus \{j\}} |k_s^i| \right) = \kappa_{ij} m_i \tag{20}$$

for all $i \in [D]$ and $j \in [N]$, where $\kappa_{ij} \geq 0$ denotes the coefficient in the bracket. $\square$

## B.2 Proof of Proposition 1

Let us estimate the distance between two output vectors of Elliptical attention mechanism corresponding to clean and contaminated query inputs, namely:

$$\boldsymbol{h} = \sum_{j \in [N]} \mathrm{softmax}(\boldsymbol{q}^\top \boldsymbol{M} \boldsymbol{k}_j / \sigma^2) \boldsymbol{v}_j = \sum_{j \in [N]} \mathcal{M}_j(\boldsymbol{q}) \boldsymbol{v}_j$$

$$\boldsymbol{h}_\epsilon = \sum_{j \in [N]} \mathrm{softmax}((\boldsymbol{q} + \boldsymbol{\epsilon})^\top \boldsymbol{M} \boldsymbol{k}_j / \sigma^2) \boldsymbol{v}_j = \sum_{j \in [N]} \mathcal{M}_j(\boldsymbol{q} + \boldsymbol{\epsilon}) \boldsymbol{v}_j,$$

where $\mathcal{M}$ is defined as in Lemma 1. We omit the keys and scaling parameter for convenience since they do not affect the analysis. Then,

$$\|\boldsymbol{h} - \boldsymbol{h}_\epsilon\| = \left\| \sum_{j\in[N]} \left(\mathcal{M}_j(\boldsymbol{q}) - \mathcal{M}_j(\boldsymbol{q}+\boldsymbol{\epsilon})\right) \boldsymbol{v}_j \right\|$$

$$\leq \sum_{j\in[N]} |\mathcal{M}_j(\boldsymbol{q}) - \mathcal{M}_j(\boldsymbol{q}+\boldsymbol{\epsilon})| \, \|\boldsymbol{v}_j\|$$

$$\leq \sum_{j\in[N]} \|\nabla\mathcal{M}_j(\hat{\boldsymbol{q}})\| \, \|\boldsymbol{\epsilon}\| \, \|\boldsymbol{v}_j\| \tag{21}$$

$$= \sum_{j\in[N]} \sqrt{\sum_{i\in[D]} (\boldsymbol{J}_\mathcal{M}(\hat{\boldsymbol{q}})_{ji})^2} \, \|\boldsymbol{v}_j\| \, \|\boldsymbol{\epsilon}\|$$

$$\leq \sum_{j\in[N]} \sqrt{\sum_{i\in[D]} \kappa_{ij}^2 m_i^2} \, \|\boldsymbol{v}_j\| \, \|\boldsymbol{\epsilon}\| \tag{22}$$

$$= \sum_{j\in[N]} \sqrt{\mathrm{tr}(\boldsymbol{K}_j^2 \boldsymbol{M}^2)} \, \|\boldsymbol{v}_j\| \, \|\boldsymbol{\epsilon}\| ,$$

where $\boldsymbol{K}_j := \mathrm{diag}(\kappa_{1j}, \kappa_{2j}, \ldots, \kappa_{Dj})$ and $\kappa_{ij}$ is defined as in Eqn. 20. Note that 21 follows from mean value theorem for some $\beta \in [0,1]$ and $\hat{\boldsymbol{q}} := \boldsymbol{q} + \beta\boldsymbol{\epsilon}$ while 22 follows from Lemma 1. $\square$

It should be noted that Proposition 1 addresses the impact of noise exclusively on the query vectors. However, the resulting bound can be extended to account for noise in all tokens by employing the same technique utilized in the proof. For completeness, we also provide the extension. Let $\mathcal{M} : \mathbb{R}^D \times \underbrace{\mathbb{R}^D \times \cdots \times \mathbb{R}^D}_{N} \to \mathbb{R}^N$ be the Elliptical Softmax function defined as

$$\mathcal{M}(\boldsymbol{q}, \boldsymbol{k}_1, \ldots, \boldsymbol{k}_N) = \frac{1}{\sum_{j\in[N]} \exp(\boldsymbol{q}^\top \boldsymbol{M} \boldsymbol{k}_j)} \begin{bmatrix} \exp(\boldsymbol{q}^\top \boldsymbol{M} \boldsymbol{k}_1) \\ \vdots \\ \exp(\boldsymbol{q}^\top \boldsymbol{M} \boldsymbol{k}_N) \end{bmatrix}. \tag{23}$$

Again, take the difference between output vectors calculated from clean and noisy tokens as follows

$$\boldsymbol{h}_\epsilon = \sum_{j\in[N]} \mathcal{M}_j(\boldsymbol{q}+\boldsymbol{\epsilon}_q, \boldsymbol{k}_1+\boldsymbol{\epsilon}_k, \ldots, \boldsymbol{k}_N+\boldsymbol{\epsilon}_k)(\boldsymbol{v}_j+\boldsymbol{\epsilon}_v), \tag{24}$$

$$\boldsymbol{h} = \sum_{j\in[N]} \mathcal{M}_j(\boldsymbol{q}, \boldsymbol{k}_1, \ldots, \boldsymbol{k}_N)\boldsymbol{v}_j. \tag{25}$$

Let $\|\bar{\boldsymbol{\epsilon}}\| := \max\{\|\boldsymbol{\epsilon}_q\|, \|\boldsymbol{\epsilon}_k\|, \|\boldsymbol{\epsilon}_v\|\}$ denote the noise with the largest norm among query, key and value noises. Then,

$$\|\boldsymbol{h}_\epsilon - \boldsymbol{h}\| \leq \sum_{j\in[N]} |\mathcal{M}_j(\boldsymbol{q}+\boldsymbol{\epsilon}_q, \boldsymbol{k}_1+\boldsymbol{\epsilon}_k, \ldots, \boldsymbol{k}_N+\boldsymbol{\epsilon}_k) - \mathcal{M}_j(\boldsymbol{q}, \boldsymbol{k}_1, \ldots, \boldsymbol{k}_N)| \|\boldsymbol{v}_j\|$$

$$+ \sum_{j\in[N]} \mathcal{M}_j(\boldsymbol{q}+\boldsymbol{\epsilon}_q, \boldsymbol{k}_1+\boldsymbol{\epsilon}_k, \ldots, \boldsymbol{k}_N+\boldsymbol{\epsilon}_k)\|\boldsymbol{\epsilon}_v\| \tag{26}$$

$$\leq \sum_{j\in[N]} \left( \|\nabla_{\boldsymbol{q}}\mathcal{M}_j(\bar{\boldsymbol{q}})\| + \sum_{s\in[N]} \|\nabla_{\boldsymbol{k}_s}\mathcal{M}_j(\bar{\boldsymbol{k}}_s)\| \right) \|\bar{\boldsymbol{\epsilon}}\| \|\boldsymbol{v}_j\| + \|\bar{\boldsymbol{\epsilon}}\|. \tag{27}$$

Following the same steps as Lemma 1, one can derive the bound

$$\left| \frac{\partial}{\partial k_s^i} \mathcal{M}_j \right| \leq m_i \cdot |q^i| \left( 1 - \frac{3\delta_{sj}}{4} \right) \propto m_i \tag{28}$$

for $s, j \in [N]$. Therefore, we obtain

$$\|\boldsymbol{h}_\epsilon - \boldsymbol{h}\| \leq \left( 1 + \sum_{j\in[N]} \sum_{s\in[N+1]} \sqrt{\mathrm{tr}(\boldsymbol{C}_{s,j}^2 \boldsymbol{M}^2)} \|v_j\| \right) \|\bar{\boldsymbol{\epsilon}}\|, \tag{29}$$

where $\boldsymbol{C}_{s,j}$ are the diagonal matrices whose elements are the proportionality coefficients in the derived upper bounds.

## B.3 Proof of Proposition 2

There are two avenues through which to see resistance to representation collapse. In this section, we provide a proof based on noise propagation through layers, which decreases representation capacity as representations in deeper layers are increasingly composed of uninformative noise. We refer the reader to Appendix B.4 for an additional lens on representation collapse, where we show that Elliptical Attention is more sensitive to the variation and local features of the underlying function.

Let the output at layer $\ell$ be denoted as $h^\ell$, the standard self-attention estimator and Elliptical estimator fitted at layer $\ell$ be denoted $\hat{f}^\ell$ and $\hat{f}^\ell_d$ respectively, where $d$ is the Mahalanobis metric described in Eqn. 7, and $f$ be the true underlying function described in Eqn. 3. By assumption, $\hat{f}$ is a higher variance estimator than $\hat{f}_d$ for any layer. The output for either estimator at layer $\ell$ can be decomposed into ground truth and noise as follows:

$$\boldsymbol{h}^\ell = \hat{f}^\ell(\boldsymbol{q}^\ell) = f(\boldsymbol{q}^\ell) + \boldsymbol{\epsilon}^\ell \tag{30}$$

$$\boldsymbol{h}^\ell_d = \hat{f}^\ell_d(\boldsymbol{q}^\ell) = f(\boldsymbol{q}^\ell) + \boldsymbol{\eta}^\ell, \tag{31}$$

where $\eta^\ell \sim \gamma(\mathbf{0}, V_\eta), \epsilon^\ell \sim \gamma(\mathbf{0}, V_\epsilon)$ are the noise components of the estimate at $\boldsymbol{q}^\ell$ and $f(\boldsymbol{q}^\ell)$ is the ground truth. By assumption of $\hat{f}_d$ being lower variance, $V_\epsilon - V_\eta$ is a positive semi-definite matrix.

We first require the following Assumption 1, which is described as:

**Assumption 1 (Random Input Noise Causes Estimator Attenuation)** . *Let $\hat{f}$ be any estimator of true function $f$ and let the input $\boldsymbol{x} \sim \mu$ drawn from marginal $\mu$ be randomly corrupted by random noise $\boldsymbol{\epsilon} \sim (0, \boldsymbol{V})$ of some unknown distribution and covariance matrix $\boldsymbol{V}$. Let $\boldsymbol{c}$ be some constant. Then, random input noise attenuates the estimator as follows:*

$$\mathbb{E}_{\boldsymbol{x},\boldsymbol{\epsilon}}\|\hat{f}(\boldsymbol{x} + \boldsymbol{\epsilon}) - \boldsymbol{c}\| \leq \mathbb{E}_{\boldsymbol{x}}\|\hat{f}(\boldsymbol{x}) - \boldsymbol{c}\| \tag{32}$$

Assumption 1 is a well-studied phenomenon in parametric regression, often referred to as attenuation bias [69], regression dilution [19], or errors-in-variables [34]. In parametric regression, it can be shown to have an exact form where the estimated gradients of the model are attenuated towards 0 proportional to the variance of the noise $\epsilon$. In non-parametric regression, addition of input noise is often referred to as random smoothing or random input smoothing [46, 10], and is well known to be used as regularization technique to introduce bias into the model. In non-parametric models, no exact closed forms exist to express the attenuation bias, but for our purposes we only note the attenuation exists and provide a general form of it in Assumption 1.

The outputs of 30 and 31 then become the inputs to the following layer after being self-added, normalized, projected, and linearly transformed. For notational simplicity and because these operations do not change the analysis, we denote the input at the next layer as the previous layer output $\boldsymbol{q}^{\ell+1} = \boldsymbol{h}^\ell$. We therefore have the following process:

$$\boldsymbol{h}^{\ell+1} = \hat{f}^{\ell+1}(\boldsymbol{q}^{\ell+1}) = \hat{f}^{\ell+1}(\boldsymbol{h}^\ell) = \hat{f}^{\ell+1}(\underbrace{f(\boldsymbol{q}^\ell) + \epsilon^\ell}_{\boldsymbol{z}^\ell}), \tag{33}$$

where we see the output $\boldsymbol{h}^{\ell+1}$ is obtained by fitting $\hat{f}^\ell$ to input $\boldsymbol{z}^\ell$ which is composed of ground truth $f(\boldsymbol{q}^\ell)$ and noise $\epsilon^\ell$ passed through from the previous layer.

The result then follows directly from the fact that in any given layer, the standard self-attention estimator produces noisier estimates, where that noise is then passed into the subsequent layer as input noise. This is

$$\mathbb{E}\|\boldsymbol{h}^{\ell+1} - \boldsymbol{c}\| = \mathbb{E}\|\hat{f}^{\ell+1}(\boldsymbol{q}^{\ell+1}) - \boldsymbol{c}\| = \mathbb{E}\|\hat{f}^{\ell+1}(f(\boldsymbol{q}^\ell) + \epsilon^\ell) - \boldsymbol{c}\| \tag{34}$$

$$\leq \mathbb{E}\|\hat{f}^{\ell+1}(f(\boldsymbol{q}^\ell) + \eta^\ell) - \boldsymbol{c}\| \tag{35}$$

$$\approx \mathbb{E}\|\hat{f}^{\ell+1}_d(f(\boldsymbol{q}^\ell) + \eta^\ell) - \boldsymbol{c}\| \tag{36}$$

$$= \mathbb{E}\|\hat{f}^{\ell+1}_d(f(\boldsymbol{q}^{\ell+1}) - \boldsymbol{c}\| = \mathbb{E}\|\boldsymbol{h}^{\ell+1}_d - \boldsymbol{c}\|, \tag{37}$$

where line 35 follows from combining the fact that $\eta^\ell$ is lower variance with Assumption 1 and line 36 follows from the fact that $\mathbb{E}\|X\| \approx \mathbb{E}\|Y\|$ when $X, Y$ have the same mean and roughly similar distribution.

Therefore we obtain at any layer $\ell$ the following

$$\mathbb{E}\|\boldsymbol{h}^{\ell+1} - \boldsymbol{c}\| \leq \mathbb{E}\|\boldsymbol{h}_d^{\ell+1} - \boldsymbol{c}\|, \tag{38}$$

as required. $\square$

### B.4 Edge-preservation Perspective on Representation Collapse

To further substantiate our findings on the mitigation of representation collapse in transformers, we now present an additional proposition that examines this phenomenon from a different perspective. In Proposition 4, we show that Elliptical attention reduces representation collapse by retaining the important local features (bumps etc.) better than the standard self-attention in the case of sparse piece-wise constant functions.

**Proposition 4 (Representation Collapse)** *Let $f : A \to \mathbb{R}^D$ for $A \subseteq \mathbb{R}^D$ be a piece-wise constant function with $f|_{A_i}(\boldsymbol{q}) = \boldsymbol{f}_i \in \mathbb{R}^D$ where $A = \bigcup_{i \in I} A_i$ for some (possibly infinite) index $I$. Let $\boldsymbol{q}_1$ and $\boldsymbol{q}_2$ be the queries lying in any of the adjacent domain pieces with distant function values. Then, the Elliptical estimates at these queries retain the distance better than the standard self-attention estimates which is formulated as*

$$\mathbb{E}\|\hat{f}_d(\boldsymbol{q}_2) - \hat{f}_d(\boldsymbol{q}_1)\| \geq \mathbb{E}\|\hat{f}(\boldsymbol{q}_2) - \hat{f}(\boldsymbol{q}_1)\|. \tag{39}$$

*Proof.* Assume all output vectors are normalized. Then, the Euclidean distance between two vectors is determined by their dot product since

$$\|\boldsymbol{a} - \boldsymbol{b}\|^2 = \|\boldsymbol{a}\|^2 + \|\boldsymbol{b}\|^2 - 2\boldsymbol{a}^\top \boldsymbol{b}. \tag{40}$$

Without loss of generality, we take $A_1$ and $A_2$ to be the two adjacent pieces so that $f(\boldsymbol{q}_i) = \boldsymbol{f}_i$ for $i = 1, 2$. Denote $\hat{f}(\boldsymbol{q}_i) = \boldsymbol{h}_i$ and $\hat{f}_d(\boldsymbol{q}_i) = \boldsymbol{h}_{id}$. Then, Eqn. 39 is equivalent to proving

$$\mathbb{E}_{\mathcal{D}}[\boldsymbol{h}_{1d}^\top \boldsymbol{h}_{2d}] \leq \mathbb{E}_{\mathcal{D}}[\boldsymbol{h}_1^\top \boldsymbol{h}_2], \tag{41}$$

where the expectation is taken over the whole sampling distribution $\mathcal{D}$ but the points $\boldsymbol{q}_1 \in A_1$ and $\boldsymbol{q}_2 \in A_2$ are fixed as described in the definition. We drop the subscript $\mathcal{D}$ as this will be the default distribution for computing expectation unless specified otherwise. Let $r_S = \text{cossim}(\boldsymbol{f}_1, \boldsymbol{f}_2) = \boldsymbol{f}_1^\top \boldsymbol{f}_2$ be the cosine similarity of the two piece-wise values. By definition of $\boldsymbol{q}_1$ and $\boldsymbol{q}_2$ and since the estimates work by averaging the output vectors with a small amount of noise, we have $r_S \leq \min\left\{\mathbb{E}[\boldsymbol{h}_{1d}^\top \boldsymbol{h}_{2d}], \mathbb{E}[\boldsymbol{h}_1^\top \boldsymbol{h}_2]\right\}$. We now decompose $\boldsymbol{h}_{1d}$ and $\boldsymbol{h}_{2d}$ in terms of components along and orthogonal to $\boldsymbol{f}_1$ and $\boldsymbol{f}_2$, respectively:

$$\boldsymbol{h}_{1d} = (\boldsymbol{h}_{1d}^\top \boldsymbol{f}_1)\boldsymbol{f}_1 + \boldsymbol{f}_1^\perp, \quad \boldsymbol{h}_{2d} = (\boldsymbol{h}_{2d}^\top \boldsymbol{f}_2)\boldsymbol{f}_2 + \boldsymbol{f}_2^\perp, \tag{42}$$

where $\boldsymbol{f}_i^\top \boldsymbol{f}_i^\perp = 0$. Then, for their dot product, we have

$$\begin{aligned}
\boldsymbol{h}_{1d}^\top \boldsymbol{h}_{2d} &= \left[(\boldsymbol{h}_{1d}^\top \boldsymbol{f}_1)\boldsymbol{f}_1 + \boldsymbol{f}_1^\perp\right]^\top \left[(\boldsymbol{h}_{2d}^\top \boldsymbol{f}_2)\boldsymbol{f}_2 + \boldsymbol{f}_2^\perp\right] \\
&= (\boldsymbol{h}_{1d}^\top \boldsymbol{f}_1)(\boldsymbol{h}_{2d}^\top \boldsymbol{f}_2)\boldsymbol{f}_1^\top \boldsymbol{f}_2 + (\boldsymbol{h}_{1d}^\top \boldsymbol{f}_1)\boldsymbol{f}_1^\top \boldsymbol{f}_2^\perp \\
&\quad + (\boldsymbol{h}_{2d}^\top \boldsymbol{f}_2)\boldsymbol{f}_2^\top \boldsymbol{f}_1^\perp + (\boldsymbol{f}_1^\perp)^\top \boldsymbol{f}_2^\perp.
\end{aligned} \tag{43}$$

The analogous decomposition of $\boldsymbol{h}_1^\top \boldsymbol{h}_2$ can be obtained. By Theorem 2 we have that the Elliptical estimator is lower variance and so we have $\mathbb{E}\|\boldsymbol{f}_i - \boldsymbol{h}_{id}\|^2 \leq \mathbb{E}\|\boldsymbol{f}_i - \boldsymbol{h}_i\|^2$. This has the following implications:

1. $1 \geq \mathbb{E}[\boldsymbol{h}_{id}^\top \boldsymbol{f}_i] \geq \mathbb{E}[\boldsymbol{h}_i^\top \boldsymbol{f}_i]$ i.e. the component of $\boldsymbol{h}_{id}$ along $\boldsymbol{f}_i$ is larger than that of $\boldsymbol{h}_i$, and hence $\boldsymbol{h}_{id}^\top \boldsymbol{f}_i$ is closer to 1.

2. Due to the first implication above, the orthogonal component $\boldsymbol{f}_i^\perp$ becomes smaller in terms of magnitude so that $\boldsymbol{f}_j^\top \boldsymbol{f}_i^\perp$ and $(\boldsymbol{f}_i^\perp)^\top \boldsymbol{f}_j^\perp$ are closer to 0 for Elliptical compared to the standard self-attention.

These two arguments, combined with Eqn. 43, imply that in expectation $\boldsymbol{h}_{1d}^\top \boldsymbol{h}_{2d}$ is closer to $1 \cdot (\boldsymbol{f}_1^\top \boldsymbol{f}_2) + 0 = \boldsymbol{f}_1^\top \boldsymbol{f}_2 = r_S$ which, by definition, is the smallest dot product over $S$, and hence, $r_S \leq \mathbb{E}[\boldsymbol{h}_{1d}^\top \boldsymbol{h}_{2d}] \leq \mathbb{E}[\boldsymbol{h}_1^\top \boldsymbol{h}_2]$ as desired. $\square$

### B.5 Proof of Proposition 3

The lemma below encapsulates the necessary calculations that will then be used in the following proofs.

**Lemma 2** *Given a normally distributed zero mean random variable $\xi \sim \mathcal{N}(0, \sigma^2)$, the expectation of a random variable obtained by its absolute value is $\mathbb{E}|\xi| = \sqrt{2\sigma^2/\pi}$.*

*Proof.* Since $\xi \sim \mathcal{N}(0, \sigma^2)$, by definition of expectation, we have

$$
\begin{aligned}
\mathbb{E}|\xi| &= \int_{-\infty}^{\infty} \frac{|x|}{\sqrt{2\pi\sigma^2}} \exp\left(-\frac{x^2}{2\sigma^2}\right) dx \\
&= \int_{-\infty}^{0} \frac{-x}{\sqrt{2\pi\sigma^2}} \exp\left(-\frac{x^2}{2\sigma^2}\right) dx + \int_{0}^{\infty} \frac{x}{\sqrt{2\pi\sigma^2}} \exp\left(-\frac{x^2}{2\sigma^2}\right) dx \qquad (44) \\
&= \frac{2}{\sqrt{2\pi\sigma^2}} \int_{0}^{\infty} x \exp\left(-\frac{x^2}{2\sigma^2}\right) dx \qquad (45) \\
&= \sqrt{\frac{2}{\pi\sigma^2}} \left[ -\sigma^2 \exp\left(-\frac{x^2}{2\sigma^2}\right) \right] \Big|_{0}^{\infty} \\
&= \sqrt{\frac{2\sigma^2}{\pi}},
\end{aligned}
$$

where we used the variable change $x \leftarrow (-x)$ in the first integral of 44 to obtain 45. $\square$

We derive the bounds for the impact of noise in 3, with respect to its variance, on our estimator 10 in Lemma 3. Henceforth, we omit the factor $\delta$ in Eqn. 10 since it does not affect the further analysis.

**Lemma 3** *Given that the noise term in 3 follows a normal distribution with zero mean and variance $\sigma^2$, the following inequality*

$$
\left| m_i - \mathbb{E}|f_i(\boldsymbol{k}^{\ell+1}) - f_i(\boldsymbol{k}^{\ell})| \right| \leq \frac{2}{\sqrt{\pi}}\sigma \qquad (46)
$$

*holds for all $i \in [D]$, where $f_i$ denotes the $i^{th}$ component of $f(\boldsymbol{k}^{\ell}) = (f_1(\boldsymbol{k}), f_2(\boldsymbol{k}), \ldots, f_D(\boldsymbol{k}))^{\top}$.*

*Proof.* Since all value vectors are taken from the data generating process 3, we have

$$
\begin{aligned}
m_i &= \mathbb{E}_{(\boldsymbol{v}^{\ell}, \boldsymbol{v}^{\ell+1}) \in \mathcal{X}_v^{\ell, \ell+1}} |\boldsymbol{v}_i^{\ell+1} - \boldsymbol{v}_i^{\ell}| \\
&= \mathbb{E}|f_i(\boldsymbol{k}^{\ell+1}) - f_i(\boldsymbol{k}^{\ell}) + \epsilon_i^{\ell+1} - \epsilon_i^{\ell}|, \qquad (47)
\end{aligned}
$$

where $\epsilon_i^{\ell}$ and $\epsilon_i^{\ell+1}$ denote the $i^{th}$ components of the noise terms $\boldsymbol{\epsilon}^{\ell}$ and $\boldsymbol{\epsilon}^{\ell+1}$, respectively. Note that for real numbers $a$ and $b$, we have by triangle inequality that $|a + b| \leq |a| + |b|$ and $|a + b| = |a - (-b)| \geq ||a| - |b|| \geq |a| - |b|$. Applying these and the linearity of expectation to 47, we obtain

$$
\mathbb{E}|f_i(\boldsymbol{k}^{\ell+1}) - f_i(\boldsymbol{k}^{\ell})| - \mathbb{E}|\epsilon_i^{\ell+1} - \epsilon_i^{\ell}| \leq m_i \leq \mathbb{E}|f_i(\boldsymbol{k}^{\ell+1}) - f_i(\boldsymbol{k}^{\ell})| + \mathbb{E}|\epsilon_i^{\ell+1} - \epsilon_i^{\ell}| \qquad (48)
$$

Recall that $\epsilon_i^{\ell} \sim \mathcal{N}(0, \sigma^2)$ and independent. Now we have that $\epsilon_i^{\ell+1} - \epsilon_i^{\ell} \sim \mathcal{N}(0, 2\sigma^2)$ as the mean value does not change while variance accumulates when subtracting two zero-mean normal variables. Therefore, the Lemma 2 gives that

$$
\mathbb{E}|\epsilon_i^{\ell+1} - \epsilon_i^{\ell}| = \frac{2}{\sqrt{\pi}}\sigma.
$$

Plugging this back into the inequalities 48, we get

$$
\mathbb{E}|f_i(\boldsymbol{k}^{\ell+1}) - f_i(\boldsymbol{k}^{\ell})| - \frac{2}{\sqrt{\pi}}\sigma \leq m_i \leq \mathbb{E}|f_i(\boldsymbol{k}^{\ell+1}) - f_i(\boldsymbol{k}^{\ell})| + \frac{2}{\sqrt{\pi}}\sigma,
$$

which is equivalent to 46 as desired. $\square$

**Remark 7** *Note that in Lemma 3, we may also take into account the possible noise in the value vectors. Let $\epsilon_{i,v}^{\ell} \sim \mathcal{N}(0, \sigma_v^2)$ be the noise in the values vectors as $m_i^{\epsilon} = \mathbb{E}|\boldsymbol{v}_i^{\ell+1} - \boldsymbol{v}_i^{\ell} + \epsilon_{i,v}^{\ell+1} - \epsilon_{i,v}^{\ell}|$. Then, applying the triangle inequality, we obtain*

$$
\mathbb{E}|\boldsymbol{v}^{\ell+1} - \boldsymbol{v}^{\ell}| - \mathbb{E}|\epsilon_{i,v}^{\ell+1} - \epsilon_{i,v}^{\ell}| \leq m_i^{\epsilon} \leq \mathbb{E}|\boldsymbol{v}^{\ell+1} - \boldsymbol{v}^{\ell}| + \mathbb{E}|\epsilon_{i,v}^{\ell+1} - \epsilon_{i,v}^{\ell}|.
$$

*Now applying Lemma 2 and Lemma 3, we arrive at*

$$
\left| m_i^{\epsilon} - \mathbb{E}|f(\boldsymbol{k}^{\ell+1}) - f(\boldsymbol{k}^{\ell})| \right| \leq \frac{2}{\sqrt{\pi}}(\sigma + \sigma_v).
$$

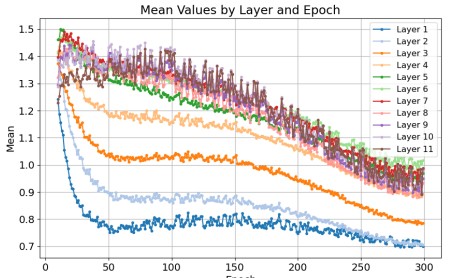 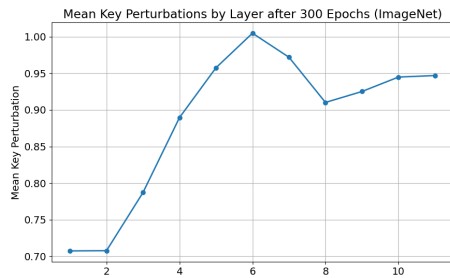

Figure 5: **Left:** Evolution of mean values of key perturbations over successive layers. **Right:** Mean key perturbations at different layers after 300 epochs. The figures show that as the number of layers increases, mean key perturbations over layers stabilize around a constant value.

*Proof of Proposition 3.* We shall first make the following assumptions on the data generating process 3:

**Assumption 2** *The underlying coordinate system in the feature space $\mathcal{X}_k$ is independent, implying that the function $f : \mathbb{R}^D \to \mathbb{R}^D$ in Eqn. 3 can be separated as $f(\boldsymbol{k}) = (f_1(k_1), \dots f_D(k_D))^\top$*

**Assumption 3** *The noise term in Eqn. 3 has independent components with each component $\epsilon_j^\ell$ following a normal distribution $\mathcal{N}(0, \sigma^2)$ for small $\sigma$, for all $j \in [D]$ and $\ell \in \mathbb{N}$*

**Assumption 4** *The magnitude of each component of key perturbations across consecutive layers, defined as $|k_i^{\ell+1} - k_i^\ell|$, follows a distribution with small, layer-independent mean ($\delta$) and variance ($\nu$)*

**Remark 8** *The assumption of layer-independence in Assumption 4, especially for deeper layers, is supported well empirically, as shown in Figure 5. Given the over-smoothing observed in transformers [24], where token representations stabilize after initial few layers, it is also practical to assume that key perturbations across layers have relatively small mean and variance when modelled as a random process.*

*Proof.* Under the Assumptions 2, 3, 4, we show that $\|f_i'\|_{1,\mu} \geq \|f_j'\|_{1,\mu}$ implies $m_i \geq m_j$ with high probability where $m_i$ is defined as in (10).

Directly from the Lemma 3, we have

$$\left| m_i - \mathbb{E}|f_i(\boldsymbol{k}^{\ell+1}) - f_i(\boldsymbol{k}^\ell)| \right| \leq \frac{2}{\sqrt{\pi}}\sigma.$$

Letting $\sigma \to 0$ in this inequality, which is feasible under the Assumption 3, one can get with a small error that

$$m_i \approx \mathbb{E}|f_i(\boldsymbol{k}^{\ell+1}) - f_i(\boldsymbol{k}^\ell)|, \tag{49}$$

which in turn implies that the impact of the noise in (10) is negligible and the error of ignoring them can be controlled by the bounds given by (46). Now according to the theorem statement,

$$\|f_i'\|_{1,\mu} \geq \|f_j'\|_{1,\mu} \iff \mathbb{E}\|\boldsymbol{J}_f(\boldsymbol{k})\boldsymbol{e}_i\|_1 \geq \mathbb{E}\|\boldsymbol{J}_f(\boldsymbol{k})\boldsymbol{e}_j\|_1$$

$$\iff \mathbb{E}\left[ \sum_{s \in [D]} \left| \frac{\partial f_s(\boldsymbol{k})}{\partial k_i} \right| \right] \geq \mathbb{E}\left[ \sum_{s \in [D]} \left| \frac{\partial f_s(\boldsymbol{k})}{\partial k_j} \right| \right]$$

$$\iff \mathbb{E}\left| f_i'(k_i) \right| \geq \mathbb{E}\left| f_j'(k_j) \right| \tag{50}$$

where we used the separability of $f$ as given in Assumption 2 which simplifies the Jacobian matrix as

$$\boldsymbol{J}_f(\boldsymbol{k}) = \begin{bmatrix} \frac{\partial f_1(\boldsymbol{k})}{\partial k_1} & \frac{\partial f_1(\boldsymbol{k})}{\partial k_2} & \cdots & \frac{\partial f_1(\boldsymbol{k})}{\partial k_D} \\ \frac{\partial f_2(\boldsymbol{k})}{\partial k_1} & \frac{\partial f_2(\boldsymbol{k})}{\partial k_2} & \cdots & \frac{\partial f_2(\boldsymbol{k})}{\partial k_D} \\ \vdots & \vdots & \ddots & \vdots \\ \frac{\partial f_D(\boldsymbol{k})}{\partial k_1} & \frac{\partial f_D(\boldsymbol{k})}{\partial k_2} & \cdots & \frac{\partial f_D(\boldsymbol{k})}{\partial k_D} \end{bmatrix} = \begin{bmatrix} \frac{\partial f_1(k_1)}{\partial k_1} & \frac{\partial f_1(k_1)}{\partial k_2} & \cdots & \frac{\partial f_1(k_1)}{\partial k_D} \\ \frac{\partial f_2(k_2)}{\partial k_1} & \frac{\partial f_2(k_2)}{\partial k_2} & \cdots & \frac{\partial f_2(k_2)}{\partial k_D} \\ \vdots & \vdots & \ddots & \vdots \\ \frac{\partial f_D(k_D)}{\partial k_1} & \frac{\partial f_D(k_D)}{\partial k_2} & \cdots & \frac{\partial f_D(k_D)}{\partial k_D} \end{bmatrix}$$

$$= \begin{bmatrix} f_1'(k_1) & 0 & \cdots & 0 \\ 0 & f_2'(k_2) & \cdots & 0 \\ \vdots & \vdots & \ddots & \vdots \\ 0 & 0 & \cdots & f_D'(k_D) \end{bmatrix},$$

so that $[\boldsymbol{J}_f(\boldsymbol{k})]_{ii} = f_i'(k_i)$. Using the definition of derivative, the inequality 50 is equivalent to

$$\mathbb{E}\left|\lim_{\tau \to 0} \frac{f^i(k_i^\ell + \tau) - f^i(k_i^\ell)}{\tau}\right| \geq \mathbb{E}\left|\lim_{\tau \to 0} \frac{f^j(k_j^\ell + \tau) - f^j(k_j^\ell)}{\tau}\right|. \tag{51}$$

Next, we note that for a small $\delta$, the limits in (51) can be approximated with $\frac{f^s(k_s^\ell + \delta) - f^s(k_s^\ell)}{\delta}$ for $s \in \{i, j\}$:

$$\frac{\mathbb{E}|f^i(k_i^\ell + \delta) - f^i(k_i^\ell)|}{\delta} \geq \frac{\mathbb{E}|f^j(k_j^\ell + \delta) - f^j(k_j^\ell)|}{\delta}. \tag{52}$$

Let us choose $\delta = \mathbb{E}|k_i^{\ell+1} - k_i^\ell|$. Then, by Chebyshev's inequality, we have for any $\varepsilon > 0$ that

$$1 - \frac{\nu^2}{\varepsilon^2} \leq \mathbb{P}\left(\left||k_i^{\ell+1} - k_i^\ell| - \delta\right| \leq \varepsilon\right)$$

$$= \mathbb{P}\left(\delta - \varepsilon \leq |k_i^{\ell+1} - k_i^\ell| \leq \delta + \varepsilon\right). \tag{53}$$

Given that the variance $\nu$ is sufficiently small as in the Assumption 4, the inequality (53) implies that $k_i^{\ell+1} \approx k_i^\ell \pm \delta$ with high probability. Therefore, it follows from (52) with high probability that

$$\frac{\mathbb{E}|f^i(k_i^{\ell+1}) - f^i(k_i^\ell)|}{\delta} \geq \frac{\mathbb{E}|f^j(k_j^{\ell+1}) - f^j(k_j^\ell)|}{\delta},$$

which, due to 49, is equivalent to $m_i \geq m_j$ as desired. $\square$

### B.6 Lipschitz smoothness in $(\mathcal{X}, d)$

Below we show how Lipschitz smoothness of $f$ changes when moving from Euclidean to the Mahalanobis transformed space. We shall follow similar steps to [29] and [30] but for a more general class of functions.

**Proposition 5 (Change in Lipschitz smoothness for $f$)** *Suppose there exists a positive constant $G_i$ such that $\|\nabla f_i(\boldsymbol{k})\| \leq G_i$ for any $\boldsymbol{k} \in \mathcal{X}_{\boldsymbol{k}}$ and $m_i > 0$ for all $i \in [D]$. Then for any $\boldsymbol{q}, \boldsymbol{k} \in \mathcal{X}_k$, the following inequality holds:*

$$\|f(\boldsymbol{q}) - f(\boldsymbol{k})\| \leq \left(\sum_{i \in [D]} \frac{G_i}{\sqrt{m_i}}\right) d(\boldsymbol{q}, \boldsymbol{k}).$$

*Proof.* Let $\boldsymbol{\omega} := \frac{\boldsymbol{q} - \boldsymbol{k}}{\|\boldsymbol{q} - \boldsymbol{k}\|}$ denote the unit vector pointing from $\boldsymbol{k}$ to $\boldsymbol{q}$. The fundamental theorem of calculus implies that

$$f(\boldsymbol{q}) - f(\boldsymbol{k}) = \int_0^{\|\boldsymbol{q} - \boldsymbol{k}\|} \frac{d}{dt} f(\boldsymbol{k} + t\boldsymbol{\omega}) \, dt = \int_0^{\|\boldsymbol{q} - \boldsymbol{k}\|} \boldsymbol{\omega}^\top \boldsymbol{J}_f(\boldsymbol{k} + t\boldsymbol{\omega}) \, dt,$$

where $\boldsymbol{J}_f$ is the Jacobian matrix of $f$ as usual. Starting with the distance between outputs $f(\boldsymbol{q})$ and $f(\boldsymbol{k})$ we have

$$\|f(\boldsymbol{q}) - f(\boldsymbol{k})\| = \left\|\int_0^{\|\boldsymbol{q} - \boldsymbol{k}\|} \boldsymbol{\omega}^\top \boldsymbol{J}_f(\boldsymbol{k} + t\boldsymbol{\omega}) \, dt\right\| \leq \int_0^{\|\boldsymbol{q} - \boldsymbol{k}\|} \left\|\sum_{i \in [D]} \omega_i \nabla f_i(\boldsymbol{k} + t\boldsymbol{\omega})\right\| dt$$

$$\leq \int_0^{\|\boldsymbol{q} - \boldsymbol{k}\|} \sum_{i \in [D]} |\omega_i| \, \|\nabla f_i(\boldsymbol{k} + t\boldsymbol{\omega})\| \, dt \leq \sum_{i \in [D]} G_i |\omega_i| \int_0^{\|\boldsymbol{q} - \boldsymbol{k}\|} dt$$

$$= \sum_{i \in [D]} G_i |q_i - k_i|, \tag{54}$$

where, as for all other vectors, $q_i$ denotes the $i^{th}$ component of vector $\boldsymbol{q}$. Now note that

$$|q_i - k_i| \leq \sqrt{(q_i - k_i)^2 + \sum_{j \neq i} \frac{m_j}{m_i}(q_j - k_j)^2} = \sqrt{\frac{(\boldsymbol{q} - \boldsymbol{k})^\top \boldsymbol{M}(\boldsymbol{q} - \boldsymbol{k})}{m_i}} = \frac{d(\boldsymbol{q}, \boldsymbol{k})}{\sqrt{m_i}}. \tag{55}$$

Combining 54 and 55, we finally attain

$$\|f(\boldsymbol{q}) - f(\boldsymbol{k})\| \leq \sum_{i \in [D]} \frac{G_i}{\sqrt{m_i}} d(\boldsymbol{q}, \boldsymbol{k}), \tag{56}$$

which completes the proof. □

## C   Additional Theorems

The following Theorem 1 is a classic result from [70]. We refer the reader to their work for details.

**Theorem 1 (Minimax rate for functions of bounded variability [70])** *Let $F_\lambda$ denote the class of distributions $P_{X,Y}$ on $\mathcal{X} \times [0,1]$ such that $\forall i \in [d]$, the directional derivates of $f(x) := \mathbb{E}[Y|X = x]$ satisfy $|f_i'|_{\sup} := \sup_{\boldsymbol{q} \in \mathcal{X}_k} \|\nabla f_i(\boldsymbol{q})\|_{\sup} \leq \lambda$. Then for any $f \in F_\lambda$, estimator $\hat{f}$, sample size $n \in \mathbb{N}$, there exists a $\tilde{c} \leq 1$ independent of $n$ satisfying*

$$\inf_{f_n} \sup_{f \in \mathcal{F}_\lambda} \mathbb{E}_{X^n, Y^n} \|\hat{f} - f\|^2 \geq 2\tilde{c}^{2/(2+d)}(d\lambda)^{2d/(2+d)} n^{-2/(2+d)} \tag{57}$$

**Theorem 2 (Improvement in MSE for approximately sparse functions [30])** *Let the norm of the largest gradient be $\lambda := \sup_{i \in [D]} \|\nabla f_i(\boldsymbol{q})\|_{\sup}$ and $\hat{f}_d$ be an estimator in metric space $(\mathcal{X}_q, d)$ where $d$ is defined as Eqn. 7. Then,*

$$\mathbb{E}\|\hat{f}_d - f\|_2^2 < \inf_{\tilde{f}} \sup_{\mathcal{F}_\lambda} \mathbb{E}\|\tilde{f} - f\|_2^2. \tag{58}$$

*Proof.* We provide an abridged proof for completeness. We refer the reader to [30] for the full details.

First, the full bound is described as follows:

$$\mathbb{E}\|\hat{f}_d - f\|_2^2 \leq 2C_{\kappa_R}^{2/2+r}(CD\lambda_d d(\mathcal{X}))^{2r/2+r} n^{-2/2+r} < \inf_{\tilde{f}} \sup_{\mathcal{F}_\lambda} \mathbb{E}\|\tilde{f} - f\|_2^2, \tag{59}$$

where $d(\mathcal{X})$ is the d-diameter of $\mathcal{X}$ defined as $\sup_{x,x' \in \mathcal{X}} d(x, x')$, $R \subset [D]$, $1 \leq C_{\kappa_R} \leq C'(4\kappa_R)^{|R|}$, $C$ and $C_1$ are universal constants and $\lambda_d \geq \sup_i \|f_i'\|_{\sup}/\sqrt{m_i}$. Let

$$r(\epsilon) \leq \begin{cases} |R| & \text{if } \epsilon \geq \epsilon_R/d(\mathcal{X}) \\ D - (D - |R|)\frac{\log(d(\mathcal{X})/\epsilon_R)}{\log(1/\epsilon)} & \text{if } \epsilon < \epsilon_R/d(\mathcal{X}) \end{cases}.$$

For bandwidth $\epsilon_n$, $r = r(\epsilon_n)$ and let $|R| \leq r \leq D$. Let $\epsilon > 0$, $\tilde{c}$ be defined as the same $\tilde{c}$ in Theorem 1, and $n \in \mathbb{N}$, define the function $\psi_{n,d} = C\epsilon^{-r(\epsilon)}/n$ and $\psi_{n,d}(\epsilon) = C_1'\epsilon^{-D}/n$ where $C_1 = \tilde{c}(\lambda/C\lambda_d d(\mathcal{X}))^D$. Also define $\phi(\epsilon) = C^2 D^2 \lambda_d^2 d(\mathcal{X})^2 \cdot \epsilon^2$.

For any fixed $n$, let $\epsilon_{n,d}$ be a solution to $\psi_{n,d}(\epsilon) = \phi(\epsilon)$. Solving for $\epsilon_{n,d}$ obtains the following lower bound on the minmax rate of

$$2\phi(\epsilon_{n,d}) = 2\tilde{c}^{2/(2+D)}(D\lambda)^{2d/(2+d)} n^{-2/(2+d)}. \tag{60}$$

For any $n \in \mathbb{N}$ there exists a solution $\epsilon_{n,d}$ to the equation $\psi_{n,d}(\epsilon) = \phi(\epsilon)$ since $r(\epsilon)$ is nondecreasing. Therefore it is possible to obtain the following:

$$\mathbb{E}_{X^n, Y^n} \|f_{n,\epsilon,d} - f\|_2^2 \leq 2\phi(\epsilon_{n,d}). \tag{61}$$

Since $\phi$ is independent of $n$, and both $\psi_{n,d}$ and $\psi_{n,d}$ are strictly decreasing functions of $n$, we have that $\epsilon_{n,d}$ and $\epsilon_{n,d}$ both tend to 0 as $n \to \infty$. Therefore we can define $n_0$ such that, for all $n \geq n_0$, both $\epsilon_{n,d}$ and $\epsilon_{n,d}$ are less than $\epsilon_R/d(\mathcal{X})$.

Thus, $\forall n \geq n_0$, we have $\epsilon_{n,d} < \epsilon_{n,d}$ if, for all $0 < \epsilon < \epsilon_R/d(\mathcal{X})$, $\psi_{n,d}(\epsilon) < \psi_{n,d}(\epsilon)$, which completes the proof □.

# D   A Consistent Estimator

In this section, we present a consistent centered difference-based quotient estimator of the coordinate-wise variability obtained by perturbing the estimated function in the $i^{th}$ direction and measuring the $L_1$ norm of the difference. Similarly, this estimator requires no learnable parameters or gradients. The estimator is described in the following proposition.

**Proposition 6 (Consistent Estimator)** *Given a function $f : \mathbb{R}^D \to \mathbb{R}^{D_v}$ with ith directional variation $\|f_i'\|_{1,\mu}, i \in [D]$, the directional variation can be estimated by the quantity*

$$\widehat{m}_i := \mathbb{E}_n \left[ \frac{\|\bar{f}(\boldsymbol{k} + t\boldsymbol{e}_i) - \bar{f}(\boldsymbol{k} - t\boldsymbol{e}_i)\|_1}{2t} \right], \tag{62}$$

*where $t$ is a hyperparameter controlling the degree of locality of the estimator and $\mathbb{E}_n$ denotes the empirical expectation for $n$ samples.*

Despite $\widehat{m}_i$ in proposition 6's simple formulation, it is nonetheless a consistent estimator of the coordinate-wise variation in the underlying function. We utilize a simplified version of a theorem from [30], adapted to suit our specific needs, as the original formulation is more detailed than necessary for our purposes.

**Theorem 3 (Consistency of Centered Difference-based Estimator for Scalar Function [30])**
*Let $\varphi : \mathbb{R}^D \to \mathbb{R}$ be a smooth scalar function and $\|\varphi_i'\|_{1,\mu} := \mathbb{E}_{\boldsymbol{x} \sim \mu} |\boldsymbol{e}_i^\top \nabla \varphi|$ be the coordinate-wise variability for that scalar function. Then, for any direction $i$ and any $0 < \delta < 1/2$, the following bound holds with probability of at least $1 - 2\delta$:*

$$\left| \mathbb{E}_n \frac{|\bar{\varphi}(\boldsymbol{x} + t\boldsymbol{e}_i) - \bar{\varphi}(\boldsymbol{x} - t\boldsymbol{e}_i)|}{2t} - \|\varphi_i'\|_{1,\mu} \right| \leq \mathcal{O}(n^{-1/2} t^{-1} \ln(2D/\delta)^{1/2}). \tag{63}$$

Note that the Theorem 3 is different from our setting by studying a scalar function as opposed to a vector valued function. However, we show that the result can be generalized to the latter case in Corollary 1 below via the estimator 62.

**Corollary 1 (Consistency of the Estimator (62) for Vector-valued Function)** *Let $f : \mathbb{R}^D \to \mathbb{R}^{D_v}$ be a vector valued function and $\|f_i'\|_{1,\mu}$ be defined as in Definition 1. Then, for any direction $i$ and any $0 < \delta < 1/2$, the following bound holds with probability of at least $1 - 2\delta$:*

$$|\widehat{m}_i - \|f_i'\|_{1,\mu}| \leq \mathcal{O}(n^{-1/2} t^{-1} \ln(2D/\delta)^{1/2}). \tag{64}$$

*Proof.* We first derive the relation between the left hand side of 64 and its coordinate-wise differences as follows:

$$|\widehat{m}_i - \|f_i'\|_{1,\mu}| = \left| \mathbb{E}_n \left[ \frac{\|\bar{f}(\boldsymbol{k} + t\boldsymbol{e}_i) - \bar{f}(\boldsymbol{k} - t\boldsymbol{e}_i)\|_1}{2t} \right] - \mathbb{E}_{\boldsymbol{k} \sim \mu} \left[ \|\boldsymbol{J}_f(\boldsymbol{k})\boldsymbol{e}_i\|_1 \right] \right|$$

$$= \left| \mathbb{E}_n \left[ \sum_{j \in [D]} \frac{|\bar{f}_j(\boldsymbol{k} + t\boldsymbol{e}_i) - \bar{f}_j(\boldsymbol{k} - t\boldsymbol{e}_i)|}{2t} \right] - \mathbb{E}_{\boldsymbol{k} \sim \mu} \left[ \sum_{j \in [D]} |\boldsymbol{e}_i^\top \nabla f_j| \right] \right| \tag{65}$$

$$= \left| \sum_{j \in [D]} \mathbb{E}_n \frac{|\bar{f}_j(\boldsymbol{k} + t\boldsymbol{e}_i) - \bar{f}_j(\boldsymbol{k} - t\boldsymbol{e}_i)|}{2t} - \sum_{j \in [D]} \mathbb{E}_{\boldsymbol{k} \sim \mu} |\boldsymbol{e}_i^\top \nabla f_j| \right| \tag{66}$$

$$= \left| \sum_{j \in [D]} \left( m_i^{(j)} - \|f_i'\|_{1,\mu}^{(j)} \right) \right| \quad \text{(definition of } m_i \text{ and } \|f_i'\|_{1,\mu} \text{ for components } f_j)$$

$$\leq \sum_{j \in [D]} \left| m_i^{(j)} - \|f_i'\|_{1,\mu}^{(j)} \right| \quad \text{(triangle inequality)}$$

$$\leq \mathcal{O}(n^{-1/2} t^{-1} \ln(2D/\delta)^{1/2}), \quad \text{(Theorem 3)}$$

where line 65 follows from the definition of the $\ell_1$ norm, line 66 follows from the linearity of expectation, the superscript $j$ indicates that the case is reduced to the scalar function case for each $j^{th}$ summand individually. Note that the probability of the last bound is at least $(1 - 2\delta/D)^D$ since each component-wise bound holds with probability at least $1 - 2\delta/D$. However, since we can choose $\delta$ small enough such that $2\delta < 1$, by Bernoulli's inequality $(1 - 2\delta/D)^D \geq 1 - 2D\delta/D = 1 - 2\delta$. $\square$

Table 9: Evaluation of the performance of our model and DeiT across multiple robustness benchmarks, using appropriate evaluation metrics for each.

| Dataset | ImageNet-R | ImageNet-A | ImageNet-C | ImageNet-C (Extra) |
| Metric | Top-1 | Top-1 | mCE ($\downarrow$) | mCE ($\downarrow$) |
| --- | --- | --- | --- | --- |
| *DeiT* | 32.22 | 7.33 | **72.21** | **63.68** |
| *DeiT-Elliptical* | **32.66** | **7.63** | 73.59 | 65.71 |

**Remark 9** *Despite the proven consistency of this estimator, we opt for the efficient estimator presented in our main body described in Eqn 10. This is because the consistent estimator requires materialising the prediction function – that is, computing a forward pass of the self-attention mechanism – twice per dimension. This makes the consistent estimator unusable in most problem settings. We present results for the consistent estimator in Appendix F.11.*

# E    Implementation Procedure and Computational Efficiency

**Training and Inference.**   Given Elliptical Attention requires keys and values from the previous layer in order to compute the required transformation, we can only implement Elliptical Attention from the second layer on. We incorporate our Elliptical Attention into both training and inference stages. This is because, firstly, Elliptical Attention is designed to offer improvements to both clean and contaminated data, and so even in the presence of completely clean train and test data, it is advantageous to incorporate Elliptical Attention into both stages. Secondly, it is commonplace to encounter data contamination in test data and indeed also highly possible to encounter it in train data as well. Therefore, in the interest of robustness as well, we also incorporate Elliptical Attention into both stages.

**Computational Efficiency.**   Computing the required transformation requires no learnable parameters and is obtained simply by averaging absolute differences in values over layers. These operations are therefore just of the order $\mathcal{O}(bhnD) = \mathcal{O}(n)$ for batch size $b$, head number $h$, key/value length $n$, and dimension $D$. Hence upper-bound time complexity of the overall Transformer is unaffected. We provide efficiency analysis in terms of computation speed and max GPU memory allocated (calculated by CUDA `max_memory_allocated` in Figure 4, which shows that compared with baseline robust models, Elliptical is the fastest and most memory efficient. Elliptical exhibits no perceptible slowdown versus DeiT of the same configuration and only a 0.99% increase in max memory allocated, which is why Elliptical and DeiT are shown as the same data point in the Figure 4.

# F    Experimental Details and Additional Experiments

## F.1    Out-of-Distribution Robustness and Data Corruption on ImageNet-A,R,C

ImageNet-A,R,C are benchmarks capturing a range of out-of-distribution and corrupted samples. ImageNet-A contains real world adversarially filtered images that fool current ImageNet classifiers. ImageNet-R contains various artistic renditions of object classes from the original ImageNet. ImageNet-C consists of 15 types of algorithmically generated corruptions with 5 levels of severity (e.g blurring, pixelation, speckle noise etc). Given that Elliptical Attention learns attention weights dependant on the transformation $M$, which is itself dependant on the train data distribution, our proposed model is not designed for situations in which the test distribution is substantially different from the train distribution. This then includes OOD robustness and robustness to heavy corruption to the point where the underlying data distribution is fundamentally different. We nonetheless evaluate Elliptical Attention on ImageNet-A,R,C to assess these important forms of robustness as well. Table 9 shows that Elliptical Attention is still able to offer improvements over baseline $DeiT$ in terms of OOD robustness, while maintaining approximately the same performance as the baseline for ImageNet-C. Figure 7 shows for *Fog* and *Pixelate* corruptions how Elliptical compares with DeiT over the 5 severity levels, where we see that at low severity levels Elliptical improves over DeiT, however as the severity level gets too high Elliptical falls behind. This agrees with our expectation that as the severity level grows, the distribution is further shifted relative to the train distribution and so Elliptical Attention is unable to improve performance.

## F.2    Representation Collapse

We provide in Figure 6 additional representation collapse results for ImageNet and ADE20K, showing that across modalities Elliptical Attention resists representation collapse.

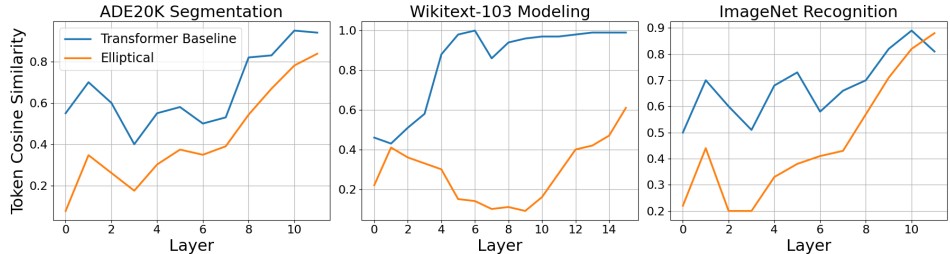

Figure 6: Additional Representation Collapse Results on ADE20K, WikiText-103 and ImageNet. Elliptical reduces token similarity over layers across a range of modalities

Table 10: Additional Results on Imagenet Increasing Heads But Maintaining Overall Embedding Dimension

| Model | Num. Heads | Head Dim. | #Params. | Top-1 Accuracy | Top-5 Accuracy |
|---|---|---|---|---|---|
| *DeiT* | 3 | 64 | 5M | 72.23 | 91.13 |
| *Elliptical* | 3 | 64 | 5M | 72.36 | 91.33 |
| *DeiT-6head* | 6 | 32 | 5M | 72.34 | 91.22 |
| *Elliptical-6head* | 6 | 32 | 5M | **73.00** | **91.77** |

## F.3   Head Redundancy

We present in Table 18 head redundancy results on the two large-scale tasks, WikiText-103 language modelling and ImageNet-1K object classification. Mean $\mathcal{L}_2$ distance between vectorized attention heads, with the mean taken over a batch of size 1000 and averaged layer-wise. We see that Elliptical improves head redundancy on WikiText-103 versus the baseline transformer while performing approxiamtely equally to the DeiT baseline on ImageNet.

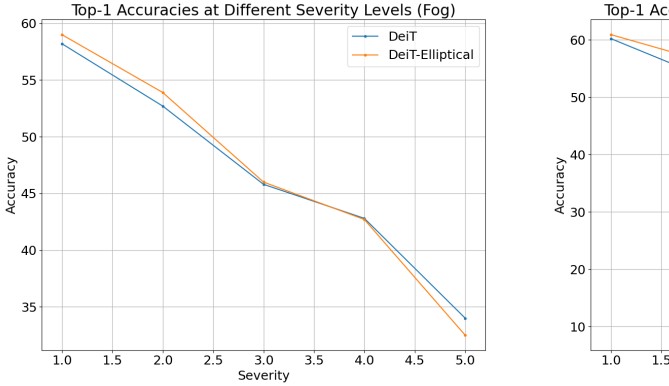
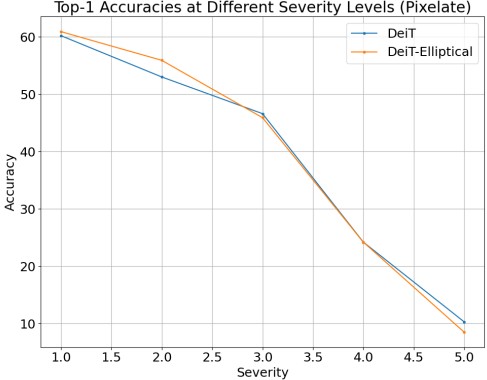

Figure 7: Comparison of *DeiT* versus *Deit-Elliptical* accuracies on two types of ImageNet-C corruptions, namely, Fog (left) and Pixelate (right). The figures show two out of many cases where *DeiT-Elliptical* outperforms its counterpart while vanilla *DeiT* manages to exceed only at higher severity levels.

## F.4   Efficiency Results

We present here the comparative efficiency results for DeiT and DeiT-Elliptical in a side-by-side comparison at tiny, small, and base sizes, along with DeiT-Elliptical compared with other robust baselines.

## F.5   Elliptical Attention in Mixture of Expert Architectures

We additionally evaluate Elliptical Attention within Mixture of Expert architectures. Specifically, we show in Tables 13, 14, and 15 the performance of Elliptical Attention within the Switch Transformer [18] and Generalized Language Model (GLaM) backbones [17].

Table 11: Side-by-side Efficiency comparison of DeiT and DeiT-Elliptical

| Model | Compute Speed (it/s) | Max Memory (K) | FLOPs / sample | #Params (M) |
|---|---|---|---|---|
| *Tiny* | | | | |
| DeiT | 0.347 | 2.08 | 1.77 | 5.7 |
| DeiT-Elliptical | 0.342 | 2.12 | 1.79 | 5.7 |
| % Change | -1.44% | 1.92% | 1.12% | - |
| *Small* | | | | |
| DeiT | 0.297 | 4.89 | 6.91 | 22.1 |
| DeiT-Elliptical | 0.289 | 4.96 | 6.99 | 22.1 |
| % Change | -2.69% | 1.43% | 1.16% | - |
| *Base* | | | | |
| DeiT | 0.132 | 10.27 | 26.37 | 86.6 |
| Deit-Elliptical | 0.130 | 10.54 | 26.63 | 86.6 |
| % Change | -1.52% | 2.63% | 0.98% | - |
| Avg. % Change | 1.88% | 1.99% | 1.09% | - |

Table 12: Efficiency Comparison between Elliptical and baseline robust models

| Model | Compute Speed (it/s) | Max Memory (K) | FLOPs / sample | #Params (M) |
|---|---|---|---|---|
| DeiT-MoM | 0.331 | 2.24 | **1.74** | 5.7 |
| DeiT-RKDE | 0.271 | 3.12 | 1.77 | 5.7 |
| DeiT-SPKDE | 0.168 | 3.35 | 1.75 | 5.7 |
| DeiT-RVT | 0.292 | 3.91 | 1.89 | 7.1 |
| DeiT-Elliptical | **0.342** | **2.12** | 1.79 | 5.7 |

## F.6 Additional Adversarial Attack Results on DeiT-Small Configuration

We present here additional results for DeiT and DeiT-Elliptical at the Small configuration [78] (22.1M parameters) under adversarial attack. Table 16 shows the result of Elliptical against PGD, FGSM, and SPSA. Table 17 shows the results of Elliptical against Auto Attack. Given the larger model size, we attack with a perturbation budget of 3/255.

## F.7 Wikitext-103 Language Modelling and Word Swap Attack

**Dataset.** The WikiText-103[2] dataset contains around 268K words and its training set consists of about 28K articles with 103M tokens. This corresponds to text blocks of about 3600 words. The validation set and test sets consist of 60 articles with 218K and 246K tokens respectively.

**Corruption.** Word Swap Text Attack[3] corrupts the data by substituting random words with a generic token 'AAA'. We follow the setup of [23] and assess models by training them on clean data before attacking only the evaluation set using a substitution rate of 2.5%.

**Model, Optimizer & Train Specification.** We adopt the training regime of [54]. To this end, the small backbone uses 16 layers, 8 heads of dimension 16, a feedforward layer of size 2048 and an embedding dimension of 128. We use a dropout rate of 0.1. We trained with Adam using a starting learning rate of 0.00025 and cosine scheduling under default PyTorch settings. We used a batch size of 96 and trained for 120 epochs and 2000 warmup steps. The train and evaluation target lengths were set to 256.

The medium backbone uses 16 layers, 8 heads of dimension 32, a feedforward layer of size 2048 and embedding dimension of 256. We use a dropout rate of 0.1. We trained with Adam using a starting

---

[2]www.salesforce.com/products/einstein/ai-research/the-wikitext-dependency-language-modeling-dataset/
[3]Implementation available at github.com/QData/TextAttack

Table 13: Elliptical Switch Transformers Pretrained on WikiText-103 and Finetuned on Stanford Sentiment Treebank 2 (SST-2)

| Model | Test PPL ($\downarrow$) | Finetune Test Acc. ($\uparrow$) |
|---|---|---|
| *Switch Transformer-medium* | 35.33 | 76.27 |
| Switch Elliptical-medium | **34.67** | **77.32** |
| *Switch Transformer-large* | 31.18 | 76.79 |
| Switch Elliptical-large | **30.56** | **78.08** |

Table 14: Elliptical Switch Transformers Pretrained on EnWik8 and Finetuned on Stanford Sentiment Treebank 2 (SST-2)

| Model | Test BPC ($\downarrow$) | Finetune Test Acc. ($\uparrow$) |
|---|---|---|
| *Switch Transformer* | 1.153 | 63.27 |
| Switch Elliptical | **1.142** | **67.75** |

learning rate 0.00025 and cosine scheduling under default PyTorch settings. We used a batch size of 56 and trained for 100 epochs and 2000 warmup steps. The train and evaluation target lengths were set to 384.

For Elliptical Attention, we use an Elliptical layer on all possible layers 2 through 16. We use a constant delta of 1.

**Compute Resources.** All models are trained and evaluated on two NVIDIA A100 SXM4 40GB GPUs.

## F.8 ImageNet Image Classification and Adversarial Attack

**Dataset.** We use the full ImageNet dataset that contains 1.28M training images and 50K validation images. The model learns to predict the class of the input image among 1000 categories. We report the top-1 and top-5 accuracy on all experiments.

**Corruption.** We use attacks FGSM [22], PGD [42], and Auto Attack [11] with perturbation budget 1/255 while SPSA [81] uses a perturbation budget 0.1. All attacks perturb under $l_\infty$ norm. PGD attack uses 20 steps with step size of 0.15.

**Model, Optimizer & Train Specification.** The configuration follows the default DeiT tiny configuration [78]. In particular, we follow the experimental setup of [23, 54]. To this end, the DeiT backbone uses 12 layers, 3 heads of dimension 64, patch size 16, feedforward layer of size 768 and embedding dimension of 192. We train using Adam with a starting learning rate of 0.0005 using cosine scheduling under default PyTorch settings, momentum of 0.9, batch size of 256, 5 warmup epochs starting from 0.000001 and 10 cooldown epochs, for an overall train run of 300 epochs. The input size is 224 and we follow the default AutoAugment policy and color jitter 0.4.

For Elliptical Attention, we use an Elliptical layer on all possible layers 2 through 12. We use a constant delta of 1.

**Compute Resources.** We train and evaluate all models on four NVIDIA A100 SXM4 40GB GPUs, with the exception of the robustness experiments on ImageNet-C which are conducted using four NVIDIA Tesla V100 SXM2 32GB GPUs.

## F.9 LRA Long Sequence Classification.

**Dataset.** The LRA benchmark consists 5 tasks involving long range contexts of up to 4000 in sequence length. These tasks consist of equation calculation (ListOps) [50], review classification (Text) [41], document retrieval (Retrieval) [61], image classification (Image) [32] and image spatial dependencies (Pathfinder) [37].

**Model, Optimizer & Train Specification.** We adopt the same experimental setup as [7]. To that end, the Transformer backbone is set with 2 layers, hidden dimension of 128, 2 attention heads

Table 15: Test Perplexity of Elliptical GLaM on WikiText-103 Modeling

| Model | Test PPL |
|---|---|
| *GLAM-small* | 58.27 |
| GLAM-Elliptical-small | **56.69** |
| *GLAM-medium* | 38.27 |
| GLAM-Elliptical-medium | **36.34** |

Table 16: DeiT and DeiT-Elliptical Accuracy on ImageNet Under Adversarial Attacks PGD, FGSM, and SPSA with Small Backbone Configuration

| Method | Clean Data | | PGD | | FGSM | | SPSA | |
|---|---|---|---|---|---|---|---|---|
| | Top 1 | Top 5 | Top 1 | Top 5 | Top 1 | Top 5 | Top 1 | Top 5 |
| *DeiT-small* | 79.89 | 95.04 | 21.41 | 51.50 | 51.57 | 82.12 | 65.68 | 91.28 |
| Elliptical-small | 79.92 | 95.06 | **22.39** | **54.02** | **51.86** | **82.87** | **72.02** | **92.45** |

Table 17: DeiT and DeiT-Elliptical Accuracy on ImageNet under Auto Attack with Small Backbone Configuration

| Method | Clean Data | | APGD-CE | | APGD-T | | FAB-T | | Square | |
|---|---|---|---|---|---|---|---|---|---|
| | Top 1 | Top 5 | Top 1 | Top 5 | Top 1 | Top 5 | Top 1 | Top 5 | Top 1 | Top 5 |
| *DeiT-small* | 79.89 | 95.04 | **19.18** | 50.75 | 16.54 | 63.84 | 80.66 | 95.09 | 49.98 | 89.17 |
| Elliptical-small | 79.92 | 95.06 | 18.88 | **51.07** | **17.30** | **65.28** | **81.64** | **95.59** | **55.89** | **89.36** |

Table 18: Head Redundancy Results

| **Model** | Num. Heads | Dim. Head | $\mathcal{L}_2$ Distance |
|---|---|---|---|
| *WikiText-103* | | | |
| *Transformer-Small* | 8 | 16 | $5.40 \pm 2.21$ |
| *Elliptical-Small* | 8 | 16 | $\mathbf{6.45} \pm 2.38$ |
| *ImageNet* | | | |
| *DeiT* | 3 | 64 | $\mathbf{5.11} \pm 1.67$ |
| *Elliptical* | 3 | 64 | $4.98 \pm 1.54$ |

of dimension 32, and embedding dimension of 64. We use a dropout rate of 0.1. Built on top of the standard transformer backbone, Reformer uses 2 hashes, Performer has 256 random feature dimensions and Linformer uses a projection dimension of 256. We train with Adam using a learning rate of 0.0001 with linear decay. We use a batch size of 32 for ListOps, Retrieval, and Text and 256 for Image and Pathfinder32. We use 1000, 175, 312, 800, and 1000 warmup steps for ListOps, Image, Pathfinder32, Retrieval, and Text respectively.

Elliptical places the Elliptical Attention layer on the final layer (as the only one possible) and uses delta equal to 1.

**Compute Resources.** All models are trained and evaluated on a single NVIDIA A100 SXM4 40GB GPU.

### F.10 ADE20K Image Segmentation

**Dataset.** ADE20K [88] contains challenging scenes with fine-grained labels and is one of the most challenging semantic segmentation datasets. The training set contains 20,210 images with 150 semantic classes. The validation and test set contain 2,000 and 3,352 images respectively.

**Model, Optimizer & Train Specification.** We follow the experimental setup as in [71]. The encoder is pretrained on ImageNet following the specification described in F.8 using the setup in [78, 54]. That is, the encoder is a DeiT backbone using 12 layers, 3 heads of dimension 64, patch

Table 19: Perplexity (PPL) of Elliptical and baselines on WikiText-103 under Word Swap data contamination. Best results are in bold. Our Elliptical method achieve substantially better robust PPL without compromising performance on clean data.

| Model | Clean Test PPL ($\downarrow$) | Contaminated Test PPL ($\downarrow$) |
|---|---|---|
| *Transformer* | 34.29 | 74.56 |
| *Performer* | 33.49 | 73.48 |
| *Transformer-MGK* | 33.21 | 71.03 |
| *FourierFormer* | 32.85 | 68.33 |
| *Transformer-SPKDE* | 32.18 | 54.97 |
| *Transformer-MoM* | 34.68 | 52.14 |
| *Elliptical* | 32.00 | 52.59 |
| *Random Ablation* | 37.84 | **46.82** |
| *Elliptical-Consistent* | 32.95 | 54.67 |
| *Elliptical-Meanscale* | **31.94** | 52.78 |

size 16, feedforward layer of size 768 and embedding dimension of 192. We train using Adam with a starting learning rate of 0.0005 using cosine scheduling under default PyTorch settings, momentum of 0.9, batch size of 256, 5 warmup epochs starting from 0.000001 and 10 cooldown epochs, for an overall train run of 300 epochs. The input size is 224 and we follow the default AutoAugment policy and color jitter 0.4. After pretraining the encoder, we then attach as decoder a masked transformer consisting of 2 layers. Each layer contains 3 heads of dimension 64, embedding dimension of 192 and feedforward dimension of 768. The decoder uses a dropout rate of 0.1. The full segmenter (encoder and decoder) is then finetuned using SGD with starting learning rate 0.001 and polynomial scheduling. The batch size is set to 8.

**Compute Resources.** All models are trained and evaluated on a single NVIDIA A100 SXM4 40GB GPU.

### F.11 Ablation Studies

**Ablation Models.** We consider the following models in our ablation studies:

- *Random Ablation.* To validate the efficacy of our proposed estimator given in Eqn. 10, we consider an alternate model in which $M$ is populated by weights uniformly drawn from the $[0, 1]$ interval followed by the same maxscaling as in *Elliptical*.

- *Elliptical-Meanscale.* We ablate the effect of maxscaling by considering meanscaling of the estimates $m_i$. That is, each $m_i \leftarrow m_i/\bar{m}$ is scaled by the mean variability estimate $\bar{m} = \mathbb{E}_D[m_i]$.

- *Elliptical-Consistent.* We consider also the performance of Elliptical when using the consistent estimator of $\|f_i'\|_{1,\mu}$ described by Equation 62.

**Language Modelling.** Results are shown in Table 19. Amazingly, the random ablation model performs extremely well on contaminated data. In general, this most likely suggests that training a model with randomness injected into the attention matrix can generate some robustness benefits, which is intuitive. It does, less surprisingly, come at the cost of clean data performance, where Random Ablation performs almost 10% worse than baseline transformer.

### F.12 Pseudocode

Algorithm 1 presents a pseudocode for implementing Elliptical Attention as given by Eqn. 13 on top of conventional self-attention.

**ImageNet Classification and Attack.** Table 21 shows the ablation model's performance on both clean ImageNet and under Auto Attack. The ablation model shows a slight improvement over the DeiT baseline in Top 1 accuracy, however Top 5 accuracy is substantially lower. Reasonable performance again Auto Attack is overall unsurprising given that the random Random Ablation model is essentially employing random defence. Nonetheless, it still does not surpass the performance of Elliptical.

**Algorithm 1** Computation of Elliptical Attention

**Require:**
1: Tensor $\boldsymbol{Q} \in \mathbb{R}^{N \times D}$ ▷ *current layer queries*
2: Tensor $\boldsymbol{K} \in \mathbb{R}^{N \times D}$ ▷ *current layer keys*
3: Tensor $\boldsymbol{V} \in \mathbb{R}^{N \times D}$ ▷ *current layer values*
4: Tensor $\boldsymbol{V}^{\text{prev}} \in \mathbb{R}^{N \times D}$ ▷ *previous layer values*
5: float $\delta \in \mathbb{R}_+$ ▷ *step size*
6: integer $D \in \mathbb{N}$ ▷ *head dimension*

7: **function** ELLIPTICAL_ATTENTION($\boldsymbol{Q}, \boldsymbol{K}, \boldsymbol{V}, \boldsymbol{V}^{\text{prev}}, \delta, D$)
8:    $\boldsymbol{M} \leftarrow$ ELLIPTICAL_WEIGHTS($\boldsymbol{V}, \boldsymbol{V}^{\text{prev}}, \delta$) ▷ *compute weight matrix $\boldsymbol{M}$*
9:    logits $\leftarrow \boldsymbol{Q} \times \boldsymbol{M} \times \boldsymbol{K}^{\top} \times \frac{1}{\sqrt{D}}$ ▷ *modify the dot-product computation*
10:   attention $\leftarrow$ SOFTMAX(logits)
11:   output $\leftarrow$ attention $\times \boldsymbol{V}$
12:   **return** output
13: **end function**

14: **function** ELLIPTICAL_WEIGHTS($\boldsymbol{V}, \boldsymbol{V}^{\text{prev}}, \delta$)
15:   **with** torch.no_grad() **do**
16:     $N \leftarrow \boldsymbol{V}$.size(0) ▷ *sequence length*
17:     value_diff $\leftarrow (\boldsymbol{V} - \boldsymbol{V}^{\text{prev}})/\delta$
18:     $\boldsymbol{M} \leftarrow \frac{1}{N} \times$ NORM(value_diff, $p = 1$, dim $= 0$) ▷ *column-wise average of $\mathcal{L}_1$ norms*
19:     $\boldsymbol{M} \leftarrow$ DIAG_EMBED($\boldsymbol{M}$) ▷ *embed the vector into a diagonal matrix*
20:   **return** $\boldsymbol{M}$
21: **end function**

Table 20: Evaluation of the performance of our model and DeiT across multiple robustness benchmarks, using appropriate evaluation metrics for each.

| Dataset
Metric | ImageNet-R
Top-1 | ImageNet-A
Top-1 | ImageNet-C
mCE ($\downarrow$) | ImageNet-C (Extra)
mCE ($\downarrow$) |
|---|---|---|---|---|
| *DeiT* | 25.38 | 3.65 | **72.21** | **63.68** |
| *Elliptical* | 31.37 | 6.76 | 73.59 | 65.71 |
| *Random Ablation* | 30.87 | 5.85 | 74.02 | 65.90 |
| *Elliptical-Consistent* | 31.46 | 6.71 | 82.92 | 71.74 |
| *Elliptical-Meanscale* | **32.66** | **7.63** | 72.28 | 63.79 |

Table 21: Auto Attack Ablation Study: Top 1 and Top 5 test accuracies on clean ImageNet and under Auto Attack. The ablation model fails to fit the clean data well and is highly prone to adversarial attack.

| Method | *DeiT* [78] | | *DeiT-Elliptical* | | *Random Ablation* | |
|---|---|---|---|---|---|---|
| | Top 1 | Top 5 | Top 1 | Top 5 | Top 1 | Top 5 |
| Clean Data | 72.23 | 91.13 | **72.36** | **91.33** | 71.44 | 91.29 |
| APGD-CE | 27.75 | 66.48 | **31.27** | **68.28** | 27.85 | 61.74 |
| APGD-T | 27.74 | 73.37 | **29.69** | **74.39** | 28.60 | 68.72 |
| FAB-T | 71.61 | 90.54 | **71.74** | **90.81** | 68.54 | 89.43 |
| Square | 43.55 | 80.96 | **47.25** | **81.65** | 47.24 | 78.87 |
| Average | 42.66 | 77.84 | **45.00** | **78.78** | 43.06 | 74.69 |
| Sequential Attack | 26.08 | 64.18 | **27.45** | **67.77** | 26.33 | 60.85 |

# G   Broader Impacts

Our research offers benefits to both clean data and robust performance. We in particular show improved results in domains with wide social applicability. These include image segmentation, with benefits to self-driving cars, and language modeling, with benefits to AI chatbot assistants. We in particular show strong improvements against contamination by adversarial attack, which we hope can protect vital AI systems from malicious actors, and competitive performance in contaminated

language modeling, which we hope can improve language models evaluated on imperfect data as is often the case in the real world. There is always possibility of misuse of AI systems, however our research shows substantive improvements in fundamental architectures and theory which we hope can spur further socially beneficial outcomes.

