# OpenReview forum: "Elliptical Attention"
_NeurIPS.cc/2024/Conference — NeurIPS 2024 poster_

### Official Review · Reviewer_5EPe · 2024-07-10

**Soundness:** 3
**Presentation:** 3
**Contribution:** 3
**Rating:** 7
**Confidence:** 3

**Summary:**

This manuscript introduces Elliptical Attention, a new approach employing the Mahalanobis distance metric to calculate attention weights. This method delineates a hyper-ellipsoidal neighborhood around each query, amplifying the attention weights of tokens situated in contextually pivotal directions. When compared with conventional self-attention mechanisms, Elliptical Attention exhibits a reduction in representation collapse and enhances the model's robustness.

**Strengths:**

1.	The paper provides a cogent blend of theoretical underpinnings and comprehensive experimental evidence supporting the efficacy of Elliptical Attention.
2.	The empirical studies conducted across diverse research benchmarks reveal that Elliptical Attention is on par with or superior to existing attention mechanisms. Notably, it consistently surpasses the baseline standard self-attention when integrated with Transformer and DeiT architectures.

**Weaknesses:**

My primary reservations pertain to the experimental framework, which necessitates a more meticulous comparative analysis. Firstly, as Elliptical Attention is posited as an alternative to traditional attention computations, its juxtaposition with the standard Euclidean distance-based self-attention should be more exhaustive. For instance, within the ImageNet-1K classification task, it would be instructive to present results for both the vanilla Vision Transformer (ViT) and a modified ViT-Elliptical, wherein the standard self-attention is supplanted by Elliptical Attention. Further empirical comparisons between DeiT and DeiT-Elliptical across various model sizes and image resolutions on the ImageNet-1K classification task would substantiate the proposed method's advantages over the conventional self-attention.

Secondly, despite assertions in the introduction regarding Elliptical Attention's reduced memory requirements and accelerated computational speed, Figure 4 shows negligible efficiency gains over the original DeiT. I would suggest the authors to furnish detailed results such as memory usage, number of model parameters, FLOPs, and throughputs for a lucid comparison. This can be done using ImageNet. They may also provide a side-by-side assessment of ViT against ViT-Elliptical.

**Questions:**

My question is to show more detailed experimental comparisons with the standard self-attention using different backbones.

**Limitations:**

The main limitation of this work is the need for a more extensive experimental comparison.

---

> ### Author Rebuttal · Authors · 2024-08-07
>
> **Q1. [Provide results for DeiT and DeiT-Elliptical at additional model sizes]**
>
> For convenience, we combine Tables E.1 and E.2 in the global response attachment into a single Table E below, which shows clean and robust performance of DeiT and DeiT-Elliptical at a larger 22.1M param model consisting of 12 layers, 384 embedding dimension over 6 heads and feed-forward size of 1536. This is the 'small' model size in [1]. We evaluate robustness using a comprehensive range of white box attacks PGD, FGSM, APGD-CE, APGD-T, and FAB-T, and black box attacks SPSA and Square. Due to the increased model size, we attack with a higher perturbation budget of 3/255.
>
> **Table E: Top-1 and Top-5 accuracy of DeiT and DeiT-Elliptical on ImageNet under Auto Attack at 22.1M Small Backbone Size**
>
>
> | Method           | Clean Data      | PGD            | FGSM            | SPSA | APGD-CE | APGD-T | FAB-T  | Square | Average |
> |------------------|-----------------|----------------|----------------|------------|---------|--------|--------|--------|---------|
> |                  | Top1/Top5   | Top1/Top5 | Top1/Top5 | Top1/Top5  | Top1/Top5  | Top1/Top5 | Top1/Top5 | Top1/Top5 | Top1/Top5 |
> | *DeiT-small*     | 79.89/95.04   | 21.41/51.50     | 51.57/82.12     | 65.68/91.28 | **19.18**/50.75 | 16.54/63.84      | 80.66/95.09      | 49.98/89.17      | 43.57/74.82     |
> | Elliptical-small | **79.92**/**95.06**   | **22.39**/**54.02**     | **51.86**/**82.87**     | **72.02/92.45** | 18.88/**51.07** | **17.30**/**65.28**  | **81.64**/**95.59**  | **55.89**/**89.36** | **45.71**/**75.80** |
>
> We see in Table E that, as with the tiny architecture, Elliptical enhances robustness across a wide variety of attacks. In particular, we see on PGD an improvement of almost 5%, and once again, Elliptical excels against black box attacks with an improvement of almost 10% and 12% in SPSA and Square, respectively.
>
> **Q2. [Show experimental comparisons of Elliptical using additional backbones]**
>
> **Answer:** We agree with the reviewer that as Elliptical Attention is posited as an alternative to traditional attention, an exhaustive comparison across a variety of backbones is key to substantiating the empirical benefits. We refer the reviewer to Tables C, D.1, and D.2 in the global response attachment, where we show the performance gains of replacing self-attention in mixture-of-expert models with Elliptical Attention. We refer additionally to Table E above as added evidence of Elliptical Attention's strong robust performance at the larger DeiT configuration size.
>
> In all, we endeavoured to provide as many tasks and backbones as possible with experimental results covering ImageNet in two different backbones and two different size configurations, language modeling and finetuning across two different model sizes covering both standard transformers and mixture-of-expert models, image segmentation, and long context analysis including CIFAR-10 object recognition, equation calculation, document retrieval, and image spatial dependency classification, all in comparison to the vanilla self-attention.  We hope that our results and these additional results offer satisfactory further evidence of our model's benefits.
>
> **Q3. [Clarify the efficiency proposition in comparison with DeiT and in comparison with the robust baselines. Provide further details on throughput, FLOPs, memory usage and parameters and compare DeiT and DeiT-Elliptical side-by-side]**
>
> **Answer:** We agree with the reviewer that there's some unintended ambiguity here that needs clarification and thank the reviewer for catching this. For the requested additional efficiency analysis, please see Table A in the global response attachment. For additional efficiency information regarding Elliptical in comparison to the other robust baselines of the same configuration, we also refer the reviewer to Table B in the global response attachment. We provide below the required clarification of the efficiency proposition made in the introduction.
>
> **Efficiency proposition clarification.** Our assertion in the introduction is that compared with comparative robust models, such as robust kernel density estimation-based approaches and Robust Vision Transformer (RVT), our model attains substantive robust performance gains at reduced memory requirements and accelerated throughput. Indeed, in Table B, we see Elliptical is both the fastest and least memory intensive. However, we do not intend to claim these efficiency benefits against the standard DeiT because, as you point out, the compute efficiency is basically the same. Essentially, our intention was to claim that the Elliptical Attention attains the efficiency gains when compared with other robust baseline models, not as in comparison to the backbone it is built atop of.
>
> In other words, in reference to Table A, we can understand the robustness gains of Elliptical Attention as coming almost for free – whatever backbone we build Elliptical Attention on top of, we do not experience significant efficiency loss as measured by throughput, memory allocation, or FLOPs. This then means that compared to other robust models (which involve computationally intensive methods for robustifying) we are far more efficient.
>
> We once again thank the reviewer for pointing out the source of confusion and we have adjusted the writing accordingly.
>
> **References**
>
> [1] Touvron, Hugo, et al. "Training data-efficient image transformers & distillation through attention." International conference on machine learning. PMLR, 2021.

---

> > ### Comment · Reviewer_5EPe · 2024-08-12
> >
> > Thanks for providing more detailed experimental comparisons in the response. My concerns are well addressed. I thus keep the accept rating.

---

> > > ### Author Response · Authors · 2024-08-12
> > > **Thanks for your endorsement!**
> > >
> > > Thanks for your response, and we appreciate your endorsement.

---

> ### Author Response · Authors · 2024-08-10
> **Any Questions from Reviewer 5EPe on Our Rebuttal?**
>
> We would like to thank the reviewer again for your thoughtful reviews and valuable feedback.
>
> We would appreciate it if you could let us know if our responses have addressed your concerns and whether you still have any other questions about our rebuttal.
>
> We would be happy to do any follow-up discussion or address any additional comments.

---

### Official Review · Reviewer_EhBA · 2024-07-11

**Soundness:** 2
**Presentation:** 2
**Contribution:** 2
**Rating:** 5
**Confidence:** 4

**Summary:**

The paper proposes a new class of self-attention mechanism for transformers. It uses Mahalanobis distance to form hyper-ellipsoidal attention regions around queries, aiming to improve model robustness and reduce representation collapse. This approach is demonstrated to be effective across various tasks like language modeling, image classification, and segmentation.

**Strengths:**

(1) The shift to hyper-ellipsoidal attention regions is a novel approach that enhances attention mechanism's sensitivity to contextually relevant features and robustness against noise.

(2) The paper provides a comprehensive set of experiments demonstrating improvements over traditional transformers in both clean and noisy conditions.

(3) It offers a theoretical framework with in-depth discussion and analysis of the proposed mechanism.

**Weaknesses:**

(1) This paper fails to provide detailed setups of experiments, making it challenging to refer to other researchers and developers.

(2) The proposed method is unclear for readers. I have not found the exact formulation of elliptical attention. Moreover, it would be helpful to include pseudo codes of core algorithms.

(3) The Fig 3 and Fig 4 are in low resolution and would be blurred after zooming in.

(4) This paper is not well written, with disconnected organization and many abbreviations.

(5) Fig 2 is pretty confusing. What does x1 and x2 mean here? What is the insight of these two subfigures? The authors failed to provide enough explanation of this figure.

**Questions:**

N/A

---

> ### Author Rebuttal · Authors · 2024-08-07
>
> **Q1. [Provide further details on experimental setup]**
>
> **Answer:** We thank the reviewer for pointing out missing details on the experimental setup, and we have added to Appendix F additional details on hyperparameters, training procedure, and optimizer specification. Papers using the same experimental setup are cited as well to help direct comparison.
>
> **Q2. [The method is unclear and the exact formulation is difficult to find. Including pseudocode would be helpful]**
>
> **Answer:**
> Thanks for your suggestion. Below we provide the requested pseudocode along with clarification on the method and formulation.
>
> **Proposed method and full technical formulation.** The proposed method replaces the Euclidean distance within self-attention with a Mahalanobis distance which stretches certain directions of the feature space in which attention is computed. These directions of the feature space correspond to contextually important directions and so by paying additional attention to tokens in those directions, the self-attention mechanism learns better representations. We discover these important directions by examining attention through the lens of non-parametric kernel regression, where we show that stretching the space in directions of *least* variability in the true underlying function obtains a lower mean squared error estimator. Adopting this approach of stretching the space from non-parametric regression into self-attention obtains our Elliptical Attention mechanism.
>
> To improve the readability of the full formulation in Section 3.5, we have updated the manuscript by repeating the definitions of the novel components immediately after providing the full formulation. We hope this helps our method be more easily understood. For reference, this is $$\boldsymbol{H} = \mathrm{softmax} \left( \frac{\boldsymbol Q\boldsymbol M\boldsymbol K^\top}{\sqrt{D}} \right) \boldsymbol V,$$
> where $\boldsymbol Q$, $\boldsymbol K$, $\boldsymbol V$ are the queries, keys, and values, respectively, $D$ is the head dimension, and $\boldsymbol M = \mathrm{diag}(m_1, m_2 \dots, m_D)$ is the diagonal Elliptical transformation with
> $$m_i := \underset{ (\boldsymbol v^\ell, \boldsymbol{v}^{\ell + 1} ) \in \mathcal{X}{v}^{\ell,\ell+1} }{\mathbb E_n} \frac{| \boldsymbol{v}^{\ell+1}(i) - \boldsymbol{v}^\ell(i) |}{\delta}$$
> where $\boldsymbol v^\ell(i)$ and $\boldsymbol v^{\ell+1}(i)$ are the $i^{th}$ dimension of values at adjacent layers $\ell$ and $\ell+1$, $\mathbb E_n$ denotes the average over $n$ values, $\mathcal{X}_{v}^{\ell,\ell+1}$ denotes the set of values across adjacent layers and $\delta$ is a hyperparameter.
>
> **Algorithm in pseudocode.** We have included an algorithm for Elliptical Attention in Appendix F.9. For reference this is:
>
> **Algorithm: Computation of Elliptical Attention**
>
> **Input:**
>
> - Tensor $Q \in \mathbb{R}^{N \times D}$ *# queries*
> - Tensor $K \in \mathbb{R}^{N \times D}$ *# keys*
> - Tensor $V \in \mathbb{R}^{N \times D}$ *# values*
> - Tensor $V^{\text{prev}} \in \mathbb{R}^{N \times D}$ *# previous layer values*
> - Float $\delta \in \mathbb{R}_{+}$ *# step size*
> - Integer $D \in \mathbb{N}$ *# head dimension*
>
> **Function:** elliptical_attention($Q$, $K$, $V$, $V^{\text{prev}}$, $\delta$, $D$)
>
> 1. $M \leftarrow$ elliptical_weights($V$, $V^{\text{prev}}$, $\delta$) *# compute weight matrix $M$*
> 2. $\text{logits} \leftarrow Q \times M \times K^\top \times \frac{1}{\sqrt{D}}$ *# modify the dot-product computation*
> 3. $\text{attention} \leftarrow \text{softmax}(\text{logits})$
> 4. $\text{output} \leftarrow \text{attention} \times V$
> 5. **return** $\text{output}$
>
> **Function:** elliptical_weights($V$, $V^{\text{prev}}$, $\delta$)
>
> 1. **with** `torch.no_grad()` **do**
>     1. $N \leftarrow V.\text{size}(0)$ *# sequence length*
>     2. value_diff $\leftarrow (V - V^{\text{prev}}) / \delta$
>     3. $M \leftarrow \frac{1}{N} \times$ norm(value_diff, $p$=1, dim=0) *# column-wise average of $\mathcal{L}_1$ norms*
>     4. $M \leftarrow$ diag_embed($M$)
> 2. **return** $M$
>
> **Q3. [Fig 3 and Fig 4 are low resolution]**
>
> **Answer:**
> We thank the reviewer for catching this and have upgraded the figures' resolution.
>
> **Q4. [Organization is disconnected and there are many abbreviations]**
>
> **Answer:**
> We endeavour to write with an intuitive, logical structure, and so we apologize if the organization was difficult to follow. If the reviewer has any specific suggestions on the organizational structure, we would be happy to implement them to improve readability.
>
> We have also updated the manuscript, rectifying undefined abbreviations.
>
>
> **Q5. [Clarify Fig 2 and its insight]**
>
> **Answer:**
> Figure 2 aims to provide a graphical explanation of the theory in Section 3.1 that distance-based estimators, such as Nadaraya-Watson, can obtain a lower mean squared error by taking ellipsoidal neighborhoods around queries. The key idea is that stretching the neighborhood in directions of least variation in the underlying function allows the model to include more points in the estimate, reducing variance without increasing bias.
>
> Specifically, Figure 2 shows a situation where $f$ (a function of two variables $x_1$ and $x_2$) does not vary equally in each direction $x_1$ and $x_2$. The left sub-figure shows the result of stretching the neighborhood around the query in the $x_2$ direction. The right sub-figure shows how this new ellipsoidal neighborhood includes 4 additional data points. This helps obtain a lower variance estimator, as more points included in the neighborhood smoothes out noise, while crucially not missing out on the true variation in $f$ because $f$ does not vary in the $x_2$ direction. Hence, we lower variance without introducing bias. This motivates our Elliptical Attention to pay more attention to keys lying in directions for which the true underlying function varies least.
>
> We have updated the writing in section 3.1 accordingly.

---

> ### Author Response · Authors · 2024-08-10
> **Any Questions from Reviewer EhBA on Our Rebuttal?**
>
> We would like to thank the reviewer again for your thoughtful reviews and valuable feedback.
>
> We would appreciate it if you could let us know if our responses have addressed your concerns and whether you still have any other questions about our rebuttal.
>
> We would be happy to do any follow-up discussion or address any additional comments.

---

> > ### Comment · Reviewer_EhBA · 2024-08-10
> > **After Rebuttal**
> >
> > Thank you for the detailed response and revisions from the authors. Points 1, 2, 3, and 5 have been well addressed. Please include the algorithm in the main manuscript to facilitate readers’ understanding of the method. I would like to increase the score to 5.

---

> > > ### Author Response · Authors · 2024-08-10
> > > **Thanks for your endorsement!**
> > >
> > > Thanks for your response, and we appreciate your endorsement. As you suggested, we will include the algorithm for Elliptical Attention in our main manuscript.

---

### Official Review · Reviewer_ySYF · 2024-07-12

**Soundness:** 3
**Presentation:** 3
**Contribution:** 3
**Rating:** 7
**Confidence:** 4

**Summary:**

This paper propose a novel attention mechanism, named Elliptical Attention.
Elliptical Attention use a Mahalanobis distance metric to stretch the underlying feature space in directions of high contextual relevance.
The Elliptical Attention pays more attention to contextually relevant information, rather than focusing on some small subset of informative  features.
The Elliptical Attention can reduce the representation collapse and enhance the model’s robustness.
This method has theoretical support and extensive experimental results on object classification, image segmentation, and language modeling validate its superiority.

This is a fundamental research.

**Strengths:**

1. This paper exhibits strong theoretical originality.
It uses a Mahalanobis distance metric to calculate the attention weights instead of the traditional pairwise dot-product.
The Mahalanobis distance metric can stretch the underlying feature space in directions of high contextual relevance.
The proposed method is well-supported theoretically.

2. The experiments are solid, which proves the effectiveness of the Elliptical Attention.
It includes object classification, image segmentation, and language modelling across different data modalities.
It analyzed the computational efficiency of this method and conducted a comprehensive research process.

3. The motivation and core ideas of the paper are articulated clearly.

**Weaknesses:**

1. The experimental table does not provide the parameter count and FLOPS.
The reviewer thinks these comparative data will provide a clearer understanding of the differences in method performance.

2. The state-of-the-art attention methods are not complete enough.
There are many improvements to classical attention mechanisms that were not compared in this paper, such as B-Attention in NeurIPS 2022. B-Attention utilizes the relationship between neighbours and improves the weight distribution of the attention mechanism.

**Questions:**

The large language model is very popular now.
The reviewer is curious whether this Elliptical Attention mechanism complies with scaling laws.
Does the Elliptical Attention have the potential to replace self-attention in LLM?

**Limitations:**

No.

---

> ### Author Rebuttal · Authors · 2024-08-07
>
> **Q1. [Provide FLOPs and parameter count]**
>
> **Answer:** We thank the reviewer for pointing out the important consideration of FLOPs and parameter count for a fuller understanding of the comparative efficiency analysis. We refer the reviewer to Tables A and B in the global response for the additional efficiency results comparing DeiT and DeiT-Elliptical side-by-side and Elliptical compared to the other robust baselines.
>
> **Q2. [Consider more SOTA methods, e.g B-Attention]**
>
> **Answer:**
> With regard to the additional SOTA methods, we first refer the reviewer to tables C, D.1, and D.2 in the global response attachment where we incorporate Elliptical into mixture-of-expert architectures and obtain substantive improvements in both language modeling pretraining and downstream finetuning tasks.
>
> With regard to B-Attention, we thank the reviewer for drawing our attention to this interesting, improved attention mechanism. This paper introduces a novel graph structure learning approach for images without inherent graph structures, employing single and multiple statistical tests (Sim-S and Sim-M) and refining the method with an efficient matrix form (B-Attention) for practical implementation. Furthermore, the high-level problem addressed in the paper – learning a robust similarity measure among high dimensional representations in the realistic setting of noisy features – is a significant and interesting perspective on our own problem setting in Elliptical Attention. The interaction between Elliptical and the B-Attention framework is indeed worth investigating, and we are currently working to integrate our mechanism into the B-Attention codebase and will present results at the earliest opportunity.
>
> Additionally, we believe it is worth mentioning that our Elliptical Attention mechanism is well suited to the problem setting of noise in features, and we demonstrate both theoretically and empirically its ability to learn a meaningful similarity metric between tokens in this noise-corrupted setting. For theoretical propositions, we refer the reviewer to Appendix B.2 where we prove that the Elliptical mechanism attenuates the effect of noise in the inputs.  Furthermore, Remark 6 in B.5 links the robustness of the  Elliptical transformation itself to noise and shows that for reasonable bounds on the data corruption, the Elliptical transformation accurately captures the contextually important directions of the feature space. Empirically, we note that our strong performance on corrupted data as shown in tables 1,2 and 4 of our experimental results (section 4) offer strong evidence that Elliptical can indeed learn an accurate similarity metric between tokens in the presence of noise.
>
> We therefore believe the theoretical background of Elliptical and B-Attention are likely to be harmonious and we will endeavour to provide additional results using this backbone as soon as possible.
>
> **Q3. [Does Elliptical comply with scaling laws and can Elliptical be integrated into LLMs?]**
>
> **Answer:**
> The reviewer raises an interesting question and direction for further research. Below, we present results that offer evidence that Elliptical Attention indeed complies with scaling laws and behaves favorably in modern LLM architectures.
>
> **Elliptical Attention and Scaling Laws.** As proposed in [1], the scaling laws for transformer language models follow the form
>
> $L(N) = (\frac{N_C}{N})^{-\alpha_N}$,
>
> where $L$ is the test loss in nats, $N$ is the number of parameters excluding the vocabulary embedding layer, and $\alpha_N$ is the degree of performance improvement expected as we scale up $N$, compute, or the amount of data used for training. $N_C$ is a constant depending on the vocabulary size and tokenization and does not have fundamental meaning. In our case, where models have limited numbers of parameters and are trained to convergence on sufficiently large datasets, the authors find $N_C \sim 8.8 \times 10^{13}$ and $\alpha_N \sim 0.076$.
>
> In our language modeling experiments, our small model has $N = 9.43M$, and our medium one has $N = 20.97M$. By the given estimates of $N_C$ and $\alpha_N$ and the scaling laws posited by the authors, this gives an anticipated relative decrease in test loss of the order
> $\frac{20.97}{9.43}^{-0.076} = 0.941$. Empirically, we find our test loss reduces by a factor $\frac{3.31782}{3.46574} = 0.957$. Hence, we find our empirical scaling results in our Elliptical language model to match almost exactly (with a margin of less than 2% difference) the existing scaling laws [1], justifying the promise of our Elliptical language model to attain the impressive and sustained improvements of LLMs.
>
> **Elliptical integration into modern LLM architectures.** New and cutting-edge LLMs are increasingly leveraging Mixture-of-expert (MoE) transformer architectures, for example Switch Transformer [2], Generalist Language Model (GLaM) [3], and Mixtral [4], along with proprietary models such as Grok and Gemini. Hence, to evaluate Elliptical Attention's compatibility with LLMs, beyond the topic of scale, we have assessed Elliptical Attention's performance within MoE transformer architectures.
>
> We refer the reviewer to the global response, where we discuss the integration of Elliptical Attention into Switch Transformer and GLaM, as well as the empirical results and strong pretraining and finetuning improvements from the resultant models, in Tables C, D.1, and D.2.
>
> **References**
>
> [1] Kaplan, Jared, et al. "Scaling laws for neural language models." (2020).
>
> [2] Fedus, William, Barret Zoph, and Noam Shazeer. "Switch transformers: Scaling to trillion parameter models with simple and efficient sparsity." 2022
>
> [3] Du, Nan, et al. "Glam: Efficient scaling of language models with mixture-of-experts." 2022
>
> [4] Jiang, Albert Q., et al. "Mixtral of experts." (2024).

---

> > ### Comment · Reviewer_ySYF · 2024-08-12
> > **Response**
> >
> > Thank you for your detailed answer !
> >
> > I have read your reply and will consider it.

---

> > > ### Author Response · Authors · 2024-08-12
> > > **Thanks for your endorsement!**
> > >
> > > Thanks for your response, and we appreciate your endorsement.

---

> ### Author Response · Authors · 2024-08-10
> **Any Questions from Reviewer ySYF on Our Rebuttal?**
>
> We would like to thank the reviewer again for your thoughtful reviews and valuable feedback.
>
> We would appreciate it if you could let us know if our responses have addressed your concerns and whether you still have any other questions about our rebuttal.
>
> We would be happy to do any follow-up discussion or address any additional comments.

---

### Author Rebuttal · Authors · 2024-08-07

Dear AC and reviewers,

Thanks for your thoughtful reviews and valuable comments, which have helped us improve the paper significantly. We are encouraged by the endorsements that: 1) the proposed method of using hyper-ellipsoidal attention regions is a novel and theoretically well-supported approach to improving the performance of Transformers on clean and contaminated data (all reviewers); 2) the experimental analysis and validation is comprehensive, demonstrating improvements across a wide array of benchmarks (all reviewers); 3) the motivation and core ideas are cogent (reviewers 5EPe, ySYF) and the theoretical framework is discussed in depth (reviewer EhBA).

One of the main questions from reviewers was regarding additional efficiency analysis, in particular looking at Elliptical and DeiT side-by-side and Elliptical and the robust baselines in terms of throughput, FLOPs, memory allocation, and parameters. Another shared question was on the performance of Elliptical Attention within additional backbones. We address these questions here.

**Additional Efficiency Analysis.** We refer the reviewers to Tables A and B in the attachment where we present throughput, memory allocation, FLOPs, and parameters for DeiT and DeiT-Elliptical side-by-side and for DeiT-Elliptical against comparative robust baselines. We show that Elliptical is the fastest and most memory efficient model compared to the robust baselines. Against the DeiT backbone, we show at tiny, small, and base sizes that Elliptical Attention incurs almost no loss in efficiency.

**Elliptical Attention in Additional Backbones.** We refer the reviewers to Tables C, D.1 and D.2 where we incorporate Elliptical Attention into mixture-of-expert baseline architectures, Switch Transformer [1] and Generalist Language Model (GLaM) [2]. We show that Elliptical Attention is highly compatible with these additional backbones, achieving substantive improvements in language modeling pretraining and downstream finetuning tasks. We also include in Tables E.1 and E.2 additional adversarial robustness results at a larger configuration, where we see strong robust performance across a variety of attacks.

We hope that our rebuttal has helped to clear concerns about our work. We are glad to answer any further questions you have on our submission and we would appreciate it if we could get your further feedback at your earliest convenience.

**References**

[1] Fedus, William, Barret Zoph, and Noam Shazeer. "Switch transformers: Scaling to trillion parameter models with simple and efficient sparsity." 2022

[2] Du, Nan, et al. "Glam: Efficient scaling of language models with mixture-of-experts." 2022

---

### Decision · Program_Chairs · 2024-09-25

**Decision:**

Accept (poster)

**Comment:**

This paper introduces the use of a Mahalanobis distance metric for computing attention weights, enabling the stretching of the underlying feature space in directions of high contextual relevance. The authors define a hyper-ellipsoidal neighborhood around each query to enhance the attention weights of tokens located in contextually significant directions.

While there are potential issues, such as the lack of detailed experimental setups, unclear model formulation, and areas needing improvement in writing and organization, this paper demonstrates strong theoretical originality. The shift to hyper-ellipsoidal attention regions is particularly novel. Additionally, the experiments are extensive, covering object classification, image segmentation, and language modeling across different data modalities. All reviewers have unanimously recommended this paper for acceptance.